# Design, Synthesis and Biological Evaluation of Phenyl Urea Derivatives as IDO1 Inhibitors

**DOI:** 10.3390/molecules25061447

**Published:** 2020-03-23

**Authors:** Chuan Zhou, Fangfang Lai, Li Sheng, Xiaoguang Chen, Yan Li, Zhiqiang Feng

**Affiliations:** 1Beijing Key Laboratory of Active Substances Discovery and Druggability Evaluation, Institute of Materia Medica, Chinese Academy of Medical Sciences and Peking Union Medical College, Beijing 100050, China; zhouchuan@imm.ac.cn; 2State Key Laboratory of Bioactive Substance and Function of Natural Medicines, Institute of Materia Medica, Chinese Academy of Medical Sciences and Peking Union Medical College, Beijing 100050, China; yanli@imm.ac.cn

**Keywords:** indoleamine 2,3-dioxygenase 1, tryptophan metabolism, immunotherapeutic target, anti-tumor, IDO1 inhibitor

## Abstract

Indoleamine 2,3-dioxygenase 1 (IDO1) is a heme-containing intracellular enzyme that catalyzes the first and rate-determining step of tryptophan metabolism and is an important immunotherapeutic target for the treatment of cancer. In this study, we designed and synthesized a new series of compounds as potential IDO1 inhibitors. These compounds were then evaluated for inhibitory activity against IDO1 and tryptophan 2,3-dioxygenase (TDO). Among them, the three phenyl urea derivatives **i12, i23, i24** as showed potent IDO1 inhibition, with IC_50_ values of 0.1–0.6 μM and no compound exhibited TDO inhibitory activity. Using molecular docking, we predicted the binding mode of compound **i12** within IDO1. Compound **i12** was further investigated by determining its in vivo pharmacokinetic profile and anti-tumor efficacy. The pharmacokinetic study revealed that compound **i12** had satisfactory properties in mice, with moderate plasma clearance (22.45 mL/min/kg), acceptable half-life (11.2 h) and high oral bioavailability (87.4%). Compound **i12** orally administered at 15 mg/kg daily showed tumor growth inhibition (TGI) of 40.5% in a B16F10 subcutaneous xenograft model and 30 mg/kg daily showed TGI of 34.3% in a PAN02 subcutaneous xenograft model. In addition, the body weight of **i12**-treated mice showed no obvious reduction compared with the control group. Overall, compound **i12** is a potent lead compound for developing IDO1 inhibitors and anti-tumor agents.

## 1. Introduction

The tryptophan/kynurenine pathway plays an important role in cancer immunotherapy [1]. Activation of this pathway promotes the degradation of tryptophan and leads to the formation of kynurenine and other bioactive metabolites, such as 3-hydroxykynurenine and 3-hydroxyanthranilic acid. A local depletion of tryptophan induces T cell cycle arrest and the accumulation of tryptophan metabolites converses naïve T cells into regulatory T cells and induces T cell apoptosis. These exert a local immunosuppressive effect which can lead to tumor progression and recurrence [2,3,4].

Indoleamine 2, 3-dioxygenase 1 (IDO1, EC 1.13.11.52) catalyzes the initial and rate-limiting step in the catabolism of tryptophan along the kynurenine pathway [5]. Cancer cells and a variety of immune cells in the tumor microenvironment are shown to overexpress IDO1, which is often associated with worse response to anticancer therapies and decreased survival of cancer patients [6,7]. Inhibition of IDO1 was shown to increase the therapeutic efficacy of cancer vaccines, immune checkpoint inhibitors, or chemotherapy in multiple clinical mouse models [8,9,10]. On this basis, several IDO1 inhibitors have been developed and are currently under clinical development (Figure 1) [1].

In this study, a new series of compounds were designed based on the phenyl urea scaffold in order to search for new IDO1 inhibitors. The compounds were synthesized and their IDO1/TDO inhibitory activities were determined. The in vivo pharmacokinetic profile and anti-tumor efficacy of a potent IDO1 inhibitor were evaluated to explore its potential as an anti-tumor agent.

## 2. Results and Discussion

### 2.1. Design Strategies of the Compounds

It was disclosed that the compound **BMS-E30** showed potent IDO1 inhibitory activity (IC_50_ = 0.7 nM) in the IDO1 kynurenine assay with human IDO1/HEK 293 cells (Scheme 1) [11], however, its enzyme inhibitory activity was relatively weak with an IC_50_ of 8.569 μM (Table 1). The lack of heme-coordinating element distinguished compound **BMS-E30** from other IDO1 inhibitors in the literature. We considered that **BMS-E30** effectively inhibited IDO1 by targeting its apo-form subsequent to the disclosure of the IDO1/**BMS-978587** crystal structure (6AZV) because the structures of **BMS-E30** and **BMS-978587** were similar (Scheme 1) [12]. Considering the two flexible chains of diisobutylamino group in compound **BMS-E30**, a new series of compounds were designed by replacing the diisobutylamino group with a 3,5-dimethylpiperidinyl group for reducing the entropy loss of binding IDO1. The peripheral phenyl urea group was also modified to explore the structure-activity relationship (SAR) which was used for further optimization to obtain better IDO1 inhibitors.

### 2.2. Synthesis of Selected Compounds

In Scheme 2, nitration of 4-fluorobenzaldehyde was performed with nitric acid and sulfuric acid to produce compound **a** [13]. Under basic conditions, compound **b** was obtained through a nucleophilic aromatic substitution reaction with 3,5-dimethylpiperidine. An aldehyde group addition reaction was performed in the presence of (trifluoromethyl)trimethylsilane to obtain compound **c** [14], which was oxidized to compound **d** with Dess-Martin periodinane. Compound **e** was prepared by a Horner-Wadsworth-Emmons reaction reaction of **d** with triethyl phosphonoacetate. The double bond and nitro group were reduced simultaneously with palladium on carbon under a hydrogen atmosphere to give compound **f**. Treatment of compound **f** with a suitable isocyanate afforded compounds **g1**–**g24**, which were further subjected to hydrolysis to obtain final products **i1**–**i24**. HATU-mediated amide coupling reaction between compound **f** and a suitable *p*-substituted phenylacetic acid formed the intermediates **h1**–**h3**, which were converted into the final products **j1**–**j3** by hydrolysis reactions.

### 2.3. In Vitro Biological Evaluation

We prepared the compounds using the synthetic scheme in Section 2.2 and these compounds were screened in vitro for their IDO1 and TDO inhibitory activities. The reported IDO1 inhibitor epacadostat was used as the reference compound in this study [15].

As shown in Table 1, compounds **g1**–**g3** showed no IDO1 inhibitory activity. When the carboxyl group in compounds **i1**–**i3** was exposed by hydrolysis, IDO1 inhibitory activity appeared. The enzymatic results of compounds **g1**–**g3** and **i1**–**i3** indicated that the carboxyl group plays a critical role in the binding activity. In addition, the replacement of phenyl ring with a cyclohexyl group (compound **i4**) or an n-hexyl group (compound **i5**) resulted in loss of inhibition. This suggested that phenyl ring is important for the binding activity.

Modification of the benzene ring of compound **i1** was performed and the results are presented in Table 2. When the *ortho*-position or the *meta*-position was substituted by CH_3_ (**i6, i7**), Cl (**i8, i9**), CN (**i10, i11**) or OCH_3_ (**i13, i14**), respectively, the obtained compounds lost all IDO1 inhibitory activity. However, *p*-substituted phenyl urea derivatives (**i2, i3, i12** and **i15**) showed more potent IDO1 inhibitory activity than the unsubstituted **i1**.

The *o,p*-disubstituted phenyl urea derivatives, bearing *o,p*-difluoro (**i16**) or *o,p*-dichloro (**i17**) substituents, were also explored and had no inhibitory activities. These results demonstrated a strong preference for *para*-substitution.

At the *para*-position, substitution with other halogens such as fluorine (**i18**, 5.475 μM) and bromine (**i19**, 4.077 μM) had similar activity as chlorine (**i3**, 5.687 μM). In addition, small *para*-alkyl substituents (**i2**, 8.613 μM; **i20**, 9.975 μM) produced inhibition against IDO1, whereas substitution with an isopropyl group (**i21**) led to the loss of activity. This result suggested that the size of the *para*-substituent on the benzene ring cannot be too large. Further optimization revealed that substitution with electron withdrawing groups at the *para*-position was beneficial for the activity (**i12**, 0.331 μM; **i23**, 0.415 μM; **i24**, 0.157 μM) and the *p*-NO_2_ derivative (**i24**) was 55-fold more potent than compound **BMS-E30**. 

Finally, replacement of proximal NH of the urea with methylene was also investigated. The observation that phenylacetamide derivatives **j1, j2, j3** had no inhibitory activity was suggestive that the proximal NH was crucial to IDO1 potency. As shown in Table 1 and Table 2, none of the synthesized compounds showed inhibitory activity against TDO, which demonstrated that the phenyl urea derivatives were selective IDO1 inhibitors.

In summary, compound **i24** showed the most potent IDO1 inhibitory activity based on the SAR study, however, it contains a nitro group which is considered to be easily metabolized and toxic in vivo [16,17,18], so the second most potent IDO1 inhibitor **i12** was used as the lead compound for further study.

### 2.4. Predicted Binding Mode of Compound ***i12*** with IDO1

The molecular docking study was performed to investigate the binding mode of compound **i12** with IDO1 (6AZV) by using CDOCKER protocol integrated in Accelrys Discovery Studio [12,19]. As shown in Figure 2, the carboxylic group in compound **i12** forms hydrogen bonds with the backbone amide of Ala-264 and with His-346, which is supposed to make critical contributions to the binding since the carboxylic group is an essential pharmacophore as SAR demonstrated. 

The phenyl urea group in compound **i12** binds via edge-to-face π-interaction with Tyr126 and hydrogen bonds with Ser167, which is also important for potency and is consistent with the SAR results. The peripheral phenyl ring was placed into the hydrophobic pocket where it is suited to extend a small *para-*substituent, which explained the loss of activity of compounds bearing bulky substituents or substituents at the *ortho*- and *meta*-positions. Particularly, electron-withdrawing groups at the *para*-position of the benzene ring were beneficial to the phenyl urea group as hydrogen bond donors, providing a rationale for 100-fold improvement in IDO1 inhibitory activity observed when compound **i12** is compared to compound **i1**.

### 2.5. Pharmacokinetic Study of Compound ***i12***

Compound **i12** was further evaluated in in vivo pharmacokinetic study using male C57BL/6 mice. The plasma concentration-time profiles are shown in Figure 3 and the pharmacokinetic parameters were determined by non-compartmental analysis (Table 3). 

When intravenously administered at a dose of 3 mg/kg, compound **i12** had moderate clearance of 22.45 mL/min/kg and an extensive distribution in tissues with V_ss_ value of 15.80 L/kg. After an oral administration of 30 mg/kg, compound **i12** was absorbed with T_max_ value of 2 h and C_max_ value of 2702 ng/mL. Compound **i12** was eliminated slowly with a half-time of 11.2 h and showed reasonable oral exposure with AUC(0–∞) value of 11,523 h·ng/mL. The oral bioavailability of compound **i12** was determined to be 87.4%, supporting a further evaluation of its in vivo efficacy.

### 2.6. In Vivo Anti-Tumor Efficacy Study of Compound ***i12***

Compound **i12** was tested for its in vivo antitumor activities in melanoma and pancreatic cancer xenograft models (Figure 4 and Figure 5). In a mouse B16F10 subcutaneous xenograft model, compound **i12** orally administered at 15 mg/kg daily demonstrated obvious in vivo anti-tumor activity and showed tumor growth inhibition (TGI) of 40.5%. In addition, the body weight of **i12**-treated mice showed no significant changes compared with the control group. The classic first line anti-cancer drug cyclophosphamide (CTX) was used as the reference drug.

We also evaluated compound **i12** in a PAN02 pancreatic cancer xenograft model, in which compound **i12** oral treatment resulted in a 34.3% decrease in tumor weight at a dose of 30 mg/kg daily compared with the control group.

## 3. Experimental Section

### 3.1. General Information

Reagents and solvents were obtained from commercial suppliers and used as received. ^1^H-NMR spectra were obtained on a 400 MHz Mercury NMR spectrometer (Varian, San Diego, CA, USA). Electrospray ionization (ESI) mass spectra and high-resolution mass spectroscopy (HRMS) were performed with a liquid chromatograph/mass selective detector time-of-flight mass spectrometer (LC/MSD TOF, Agilent Technologies, Santa Clara, CA, USA). Silica gel column chromatography was performed with silica gel 60G (Qingdao Haiyang Chemical, Qingdao, China). Purity was determined using HPLC, LC/MS and NMR spectroscopy (Appendix A). All of the synthesized compounds have purities over 95%.

#### 3.1.1. Preparation of 4-Fluoro-3-nitro-benzaldehyde (**a**)

To a solution of sulfuric acid (24 mL) and nitric acid (3 mL) was slowly added 4-fluoro-benzaldehyde (6 g, 48.3 mmol) at −5 °C. The reaction mixture was stirred at room temperature for 2 h, poured onto ice and extracted with ethyl acetate (200 mL). The organic extract was washed with brine (60 mL × 2), dried over anhydrous Na_2_SO_4_ and concentrated. The residue was purified by column chromatography (silica gel, PE/EA = 30:1, *v*/*v*) to afford compound a as a light yellow solid (7.8 g, 95.4% yield). ^1^H-NMR (CDCl_3_): δ 10.05 (s, 1H, CHO), 8.60 (dd, *J* = 7.0, 1.7 Hz, 1H, H-phenyl), 8.21 (ddd, *J* = 8.6, 4.2, 2.1 Hz, 1H, H-phenyl), 7.51 (dd, *J* = 9.9, 8.7 Hz, 1H, H-phenyl). HRMS (ESI) *m*/*z*: [M + H]^+^ calculated for C_7_H_5_O_3_NF, 170.02480; found, 170.02446, Δ −1.99 ppm.

#### 3.1.2. Preparation of 4-(3,5-Dimethylpiperidin-1-yl)-3-nitro-benzaldehyde (**b**)

To a solution of 4-fluoro-3-nitrobenzaldehyde (**a**, 5.9 g, 34.9 mmol) in dichloromethane (100 mL) was added 3,5-dimethylpiperidine (5.9 g, 52.1 mmol) and ethylamine (5.3 g, 52.4 mmol) at 0 °C. The resulting mixture was stirred at room temperature for 0.5 h, poured into water (200 mL) and extracted with dichloromethane (200 mL × 2). The combined organic layers were washed with 1N HCl aqueous solution (150 mL × 2), saturated NaCl aqueous solution (150 mL × 2), dried over anhydrous Na_2_SO_4_ and concentrated. The residue was purified by column chromatography (silica gel, PE/EA = 10:1, *v*/*v*) to afford the product b as a yellow solid (8.6 g, 93.5% yield). ^1^H-NMR (DMSO-*d*_6_): δ 9.82 (s, 1H, CHO), 8.31 (s, 1H, H-phenyl), 7.90 (t, *J* = 9.8 Hz, 1H, H-phenyl), 7.39 (d, *J* = 8.6 Hz, 1H, 1H-phenyl), 3.30 (d, *J* = 12.9 Hz, 2H, CH_a_-1 and CH_a_-5), 2.58 (t, *J* = 12.0 Hz, 2H, CH_b_-1 and CH_b_-5), 1.84–1.65 (m, 3H, CH_a_-3, CH-2, CH-4), 0.92–0.74 (m, 8H, CH_3_-6, CH_3_-7 and CH_b_-3). HRMS (ESI) *m*/*z*: [M + H]^+^ calculated for C_14_H_19_O_3_N_2_, 263.13902; found, 263.13876, Δ −0.98 ppm.

#### 3.1.3. Preparation of 1-[4-(3,5-Dimethylpiperidin-1-yl)-3-nitro-phenyl]-2,2,2-trifluoroethanol (**c**)

To a solution of 4-(3,5-dimethylpiperidin-1-yl)-3-nitrobenzaldehyde (**b**, 4.2 g, 16.0 mmol) and trimethyl(trifluoromethyl)silane (4.6 g, 32.0 mmol) in dimethylformamide (40 mL) was added potassium carbonate (4.4 g, 32.0 mmol) at 0 °C. The resulting mixture was warmed to room temperature and stirred for 12 h. The mixture was then treated with 40 mL of 1 mol/L HCl aqueous solution, stirred for another 30 min, poured into water (200 mL) and extracted with ethyl acetate (200 mL × 2). The organic extract was washed with saturated NaCl aqueous solution (150 mL × 3), dried over anhydrous Na_2_SO_4_ and concentrated. The residue was purified by column chromatography (silica gel, PE/EA = 8:1, *v*/*v*) to afford the product c as a brown red oil (4.6 g, 86.8% yield). ^1^H-NMR (DMSO-*d*_6_): δ 7.91 (d, *J* = 1.9 Hz, 1H, H-phenyl), 7.64 (dd, *J* = 8.6, 1.7 Hz, 1H, H-phenyl), 7.31 (dd, *J* = 8.7, 2.5 Hz, 1H, H-phenyl), 7.00–6.94 (m, 1H, CH), 5.24 (d, *J* = 7.0 Hz, 1H, OH), 3.13 (dd, *J* = 12.0, 1.5 Hz, 2H, CH_a_-1 and CH_a_-5), 2.38 (t, *J* = 11.5 Hz, 2H, CH_b_-1 and CH_b_-5), 1.81–1.65 (m, 3H, CH_a_-3, CH-2, CH-4), 0.85 (d, *J* = 6.5 Hz, 6H, CH_3_-6, CH_3_-7), 0.69 (q, *J* = 9.6 Hz, 1H, CH_b_-3). HRMS (ESI) *m*/*z*: [M + H]^+^ calculated for C_15_H_20_O_3_N_2_F_3_, 333.14205; found, 333.14246, Δ 1.22 ppm.

#### 3.1.4. Preparation of 3-[4-(3,5-Dimethylpiperidin-1-yl)-3-nitrophenyl]-4,4,4-trifluorobut-2-enoic Acid Ethyl Ester (**e**)

To a solution of 1-[4-(3,5-dimethylpiperidin-1-yl)-3-nitrophenyl]-2,2,2-trifluoroethanol (**c**, 4.2 g, 12.6 mmol) in dichloromethane (200 mL) at 0 °C was added sodium bicarbonate (3.2 g, 37.8 mmol) followed by Dess-Martin periodinane (8.0 g, 18.9 mmol). The resulting solution was warmed to room temperature and stirred for 2 h. The reaction mixture was then diluted with saturated NaHCO_3_ aqueous solution (100 mL) and stirred for another 30 min. The organic layer was separated and washed with saturated NaCl aqueous solution (100 mL × 2), dried over anhydrous Na_2_SO_4_ and concentrated. The residue was preliminarily purified by column chromatography (silica gel, PE/EA = 20:1, *v*/*v*) to afford crude product **d** as a yellow solid. The crude product **d** was used directly without further purification. Next ethyl 2-(diethoxyphosphoryl)acetate (1.1 g, 4.9 mmol) was added to a solution of NaH (2.28 mg, 5.7 mmol) in dry tetrahydrofuran at 0 °C. The reaction mixture became a clear solution after stirring for 30 min. The crude product **d** in dry tetrahydrofuran was then added. The resulting mixture was warmed to room temperature, stirred for 2 h, quenched with saturated NH_4_Cl aqueous solution (10 mL). The aqueous layer was further extracted with ethyl acetate (50 mL × 2) and the combined organic extracts were washed with saturated NaCl aqueous solution (50 mL × 2), dried over anhydrous Na_2_SO_4_ and concentrated. The residue was purified by column chromatography (silica gel, PE/EA = 40:1, *v*/*v*) to afford the product e as an orange yellow oil (1.64 g, 32.5% yield). ^1^H-NMR (DMSO-*d*_6_): δ 7.73 (d, *J* = 2.2 Hz, 1H, H-phenyl), 7.44 (dd, *J* = 8.7, 2.0 Hz, 1H, H-phenyl), 7.32 (dd, *J* = 8.8, 2.8 Hz, 1H, H-phenyl), 6.86 (s, 1H, =CH), 4.04 (q, *J* = 7.2 Hz, 2H, CH_2_), 3.21–3.13 (m, 2H, CH_a_-1 and CH_a_-5), 2.44 (t, *J* = 11.7 Hz, 2H, CH_b_-1 and CH_b_-5), 1.83–1.64 (m, 3H, CH_a_-3, CH-2, CH-4), 1.04 (t, *J* = 7.2 Hz, 3H, CH_3_), 0.85 (d, *J* = 6.5 Hz, 6H, CH_3_-6, CH_3_-7), 0.73 (q, *J* = 11.8 Hz, 1H, CH_b_-3). HRMS (ESI) *m*/*z*: [M + H]^+^ calculated for C_15_H_20_O_3_N_2_F_3_, 401.16827; found, 401.16788, Δ −0.97 ppm.

#### 3.1.5. Preparation of 3-[3-Amino-4-(3,5-dimethylpiperidin-1-yl)-phenyl]-4,4,4-trifluorobutyric Acid Ethyl Ester (**f**)

To a solution of 3-[4-(3,5-dimethylpiperidin-1-yl)-3-nitrophenyl]-4,4,4-trifluorobut-2-enoic acid ethyl ester (**e**, 2.4 g, 6.0 mmol) in dry tetrahydrofuran (60 mL) was added palladium on carbon (640 mg, 10% Pd/C, 0.6 mmol) and the suspension was hydrogenated (1 atm, balloon) for 2 h. Thin layer chromatography (TLC) indicated completion. The suspension was filtered through a pad of Celite and the filter cake was rinsed with ethyl acetate (50 mL × 3). The combined filtrate and rinses were concentrated and the residue was purified by column chromatography (silica gel, PE/EA = 30:1, *v*/*v*) to afford the product f as a colorless oil (945 mg, 42.3% yield). ^1^H-NMR (DMSO-*d*_6_): δ 6.82 (d, *J* = 8.1 Hz, 1H, H-phenyl), 6.67 (d, *J* = 1.7 Hz, 1H, H-phenyl), 6.54 (dd, *J* = 8.0, 1.8 Hz, 1H, H-phenyl), 4.76 (s, 2H, NH_2_), 4.07–3.93 (m, 2H, CH_2_), 3.85–3.70 (m, 1H, CH), 3.02–2.76 (m, 4H, CH_2_, CH_a_-1 and CH_a_-5), 2.00 (td, *J* = 10.9, 4.4 Hz, 2H, CH_b_-1 and CH_b_-5), 1.86–1.70 (m, 3H, CH_a_-3, CH-2, CH-4), 1.08 (t, *J* = 7.1 Hz, 3H, CH_3_), 0.85 (d, *J* = 6.4 Hz, 6H, CH_3_-6, CH_3_-7), 0.68–0.56 (m, 1H, CH_b_-3). HRMS (ESI) *m*/*z*: [M + H]^+^ calculated for C15H20O3N2F3, 373.20974; found, 373.20999, Δ 0.67 ppm.

#### 3.1.6. General Procedure A for the Synthesis of **g1**–**g24**

To a solution of 3-[3-amino-4-(3,5-dimethylpiperidin-1-yl)-phenyl]-4,4,4-trifluorobutyric acid ethyl ester (**f**, 1 equiv.) in tetrahydrofuran was added a suitable isocyanate (1 equiv.). The reaction mixture was stirred at room temperature until the starting material disappeared in TLC. The reaction mixture was concentrated and the residue was purified by column chromatography (silica gel, PE/EA = 8:1, *v*/*v*) to afford the products **g1**–**g24**.

#### 3.1.7. General Procedure B for the Synthesis of **i1–i24** and **j1–j3**

To a solution of compound **g1–g24** or **h1–h3** (1 equiv.) in tetrahydrofuran (4 volumes), methanol (1 volume) and water (1 volume) was added sodium hydroxide (3 equiv.). The resulting mixture was stirred at room temperature until the starting material disappeared in TLC. Part of the tetrahydrofuran and methanol was removed in vacuo and the crude was diluted with water (2 volumes) and the pH was adjusted to ca. 4 using 1 N HCl solution. The aqueous phase was then extracted with ethyl acetate (15 v × 3) and the combined organic extracts were washed with saturated NaCl solution, dried over anhydrous Na_2_SO_4_ and concentrated. The residue was purified by column chromatography (silica gel, DCM/MeOH = 30:1, *v*/*v*) to afford the products **i1–i24** or **j1–j3**.

#### 3.1.8. Preparation of Racemic 3-[4-(3,5-dimethylpiperidin-1-yl)-3-(3-phenylureido)-phenyl]-4,4,4-trifluorobutyric Acid Ethyl Ester (**g1**)

Reaction of compound **f** and phenyl isocyanate following the general procedure A afforded compound **g1** (white solid, 85.3% yield). ^1^H-NMR (DMSO-*d*_6_): δ 9.54 (s, 1H, NH), 8.14 (d, *J* = 1.6 Hz, 1H, H-phenyl), 8.07 (s, 1H, NH), 7.49 (d, *J* = 7.9 Hz, 2H, H-phenyl), 7.29 (t, *J* = 7.8 Hz, 2H, H-phenyl), 7.13 (d, *J* = 8.2 Hz, 1H, H-phenyl), 7.02–6.95 (m, 2H, H-phenyl), 4.07–3.87 (m, 3H, CH_2_, CH), 3.07–2.79 (m, 4H, CH_2_, CH_a_-1 and CH_a_-5), 2.14 (t, *J* = 10.3 Hz, 2H, CH_b_-1 and CH_b_-5), 2.06–1.91 (m, 2H, CH-2, CH-4), 1.80 (d, *J* = 12.7 Hz, 1H, CH_a_-3), 1.08 (t, *J* = 7.1 Hz, 3H, CH_3_), 0.87 (d, *J* = 6.6 Hz, 6H, CH_3_-6, CH_3_-7), 0.67 (q, *J* = 12.0 Hz, 1H, CH_b_-3). ^13^C-NMR (DMSO-*d*_6_): δ 169.61, 152.54, 141.92, 139.82, 134.19, 129.17, 128.87 (2C), 128.14, 122.53, 122.04, 120.38, 119.34, 118.56 (2C), 60.48, 59.72 (2C), 45.02, 41.62, 33.81, 30.87 (2C), 19.33 (2C), 13.92. HRMS (ESI) *m*/*z*: [M + H]^+^ calculated for C_26_H_33_O_3_N_3_F_3_, 492.24685; found, 492.24582, Δ −2.10 ppm.

#### 3.1.9. Preparation of Racemic 3-[4-(3,5-dimethylpiperidin-1-yl)-3-(3-phenylureido)-phenyl]-4,4,4-trifluorobutyric Acid (**i1**)

Hydrolysis of compound **g1** (80 mg, 0.16 mmol) following the general procedure B afforded compound **h1** (white solid, 61 mg, 80.7% yield). mp: 94.1–95.7 °C. ^1^H-NMR (DMSO-*d*_6_): δ 12.61 (s, 1H, COOH), 9.58 (s, 1H, NH), 8.14 (d, *J =* 1.9 Hz, 1H, H-phenyl), 8.09 (s, 1H, NH), 7.49 (dd, *J =* 8.6, 1.0 Hz, 2H, H-phenyl), 7.33–7.26 (m, 2H, H-phenyl), 7.13 (d, *J =* 8.2 Hz, 1H, H-phenyl), 7.02–6.95 (m, 2H, H-phenyl), 3.95–3.83 (m, 1H, CH), 2.97–2.71 (m, 4H, CH_2_, CH_a_-1 and CH_a_-5), 2.14 (td, *J =* 11.2, 3.7 Hz, 2H, CH_b_-1 and CH_b_-5), 2.05–1.91 (m, 2H, CH-2, CH-4), 1.80 (d, *J =* 12.8 Hz, 1H, CH_a_-3), 0.87 (d, *J =* 6.6 Hz, 6H, CH_3_-6, CH_3_-7), 0.66 (q, *J =* 12.4 Hz, 1H, CH_b_-3). HRMS(ESI) *m*/*z*: [M + H]^+^ calculated for C_24_H_29_O_3_N_3_F_3_, 464.21555; found, 464.21466, Δ −1.92 ppm.

#### 3.1.10. Preparation of Racemic 3-[4-(3,5-dimethylpiperidin-1-yl)-3-(3-*p*-tolylureido)phenyl]-4,4,4-trifluorobutyric Acid Ethyl Ester (**g2**)

Reaction of compound **f** and *p*-tolyl isocyanate following the general procedure A afforded compound **g2** (white solid, 86.1% yield). ^1^H-NMR (DMSO-*d*_6_): δ 9.42 (s, 1H, NH), 8.14 (d, *J* = 2.0 Hz, 1H, H-phenyl), 8.02 (s, 1H, NH), 7.38–7.35 (m, 2H, H-phenyl), 7.15–7.07 (m, 3H, H-phenyl), 6.97 (dd, *J* = 8.2, 2.0 Hz, 1H, H-phenyl), 4.07–3.88 (m, 3H, CH_2_, CH), 3.06–2.79 (m, 4H, CH_2_, CH_a_-1 and CH_a_-5), 2.25 (s, 3H, CH_3_-phenyl), 2.13 (td, *J* = 11.2, 2.0 Hz, 2H, CH_b_-1 and CH_b_-5), 2.02–1.90 (m, 2H, CH-2, CH-4), 1.79 (d, *J* = 13.1 Hz, 1H, CH_a_-3), 1.07 (t, *J* = 6.8 Hz, 3H, CH_3_), 0.86 (d, *J* = 6.6 Hz, 6H, CH_3_-6, CH_3_-7), 0.66 (q, *J* = 12.4 Hz, 1H, CH_b_-3). ^13^C-NMR (DMSO-*d*_6_): δ 169.61, 152.59, 141.81, 137.19, 134.32, 130.95, 129.28 (2C), 129.20, 122.40, 120.39, 119.19, 118.79 (2C), 118.27, 60.48, 59.71 (2C), 44.90, 41.62, 33.81, 30.87 (2C), 20.42, 19.33 (2C), 13.92. HRMS (ESI) *m*/*z*: [M + H]^+^ calculated for C_27_H_35_O_3_N_3_F_3_, 506.26250; found, 506.26242, Δ −0.16 ppm.

#### 3.1.11. Preparation of Racemic 3-[4-(3,5-dimethylpiperidin-1-yl)-3-(3-*p*-tolylureido)phenyl]-4,4,4-trifluorobutyric Acid (**i2**)

Hydrolysis of compound **g2** (140 mg, 0.28 mmol) following the general procedure B afforded compound **i2** (white solid, 113 mg, 85.4% yield). mp: 134.3–136.1 °C. ^1^H-NMR (DMSO-*d*_6_): δ 12.50 (s, 1H, COOH), 9.45 (s, 1H, NH), 8.15 (d, *J =* 1.8 Hz, 1H, H-phenyl), 8.04 (s, 1H, NH), 7.37 (d, *J =* 8.4 Hz, 2H, ArH-phenyl), 7.15–7.06 (m, 3H, H-phenyl), 6.97 (dd, *J =* 8.2, 1.8 Hz, 1H, H-phenyl), 3.96–3.82 (m, 1H, CH), 2.99–2.75 (m, 4H, CH_2_, CH_a_-1 and CH_a_-5), 2.25 (s, 3H, -CH_3_, CH_3_-phenyl), 2.13 (td, *J =* 11.1, 3.3 Hz, 2H, CH_b_-1 and CH_b_-5), 2.03–1.89 (m, 2H, CH-2, CH-4), 1.79 (d, *J =* 12.7 Hz, 1H, CH_a_-3), 0.86 (d, *J =* 6.6 Hz, 6H, CH_3_-6, CH_3_-7), 0.66 (q, *J =* 12.0 Hz, 1H, CH_b_-3). ^13^C-NMR (DMSO-*d*_6_): δ 171.05, 152.62, 141.76, 137.22, 134.36, 130.95, 129.48, 129.28 (2C), 126.88, 122.49, 120.39, 119.17, 118.82 (2C), 59.75 (2C), 45.14, 41.65, 33.80, 30.87 (2C), 20.42, 19.33 (2C). HRMS (ESI) *m*/*z*: [M + H]^+^ calculated for C_25_H_31_O_3_N_3_F_3_, 478.23120; found, 478.23230, Δ 2.29 ppm.

#### 3.1.12. Preparation of Racemic 3-[3-[3-(4-chlorophenyl)-ureido]-4-(3,5-dimethylpiperidin-1-yl)-phenyl]-4,4,4-trifluorobutyric Acid Ethyl Ester (**g3**)

Reaction of compound **f** and 4-chlorophenyl isocyanate following the general procedure A afforded compound **g3** (white solid, 83.4% yield). ^1^H-NMR (DMSO-*d*6): δ 9.69 (s, 1H, NH), 8.14 (d, *J =* 1.9 Hz, 1H, H-phenyl), 8.09 (s, 1H, NH), 7.55–7.49 (m, 2H, H-phenyl), 7.37–7.31 (m, 2H, H-phenyl), 7.14 (d, *J =* 8.2 Hz, 1H, H-phenyl), 7.00 (dd, *J =* 8.2, 1.9 Hz, 1H, H-phenyl), 4.07–3.88 (m, 3H, CH_2_, CH), 3.07–2.76 (m, 4H, CH_2_, CH_a_-1 and CH_a_-5), 2.14 (t, *J =* 10.6 Hz, 2H, CH_b_-1 and CH_b_-5), 2.06–1.91 (m, 2H, CH-2, CH-4), 1.80 (d, *J =* 12.9 Hz, 1H, CH_a_-3), 1.07 (t, *J =* 7.2 Hz, 3H, CH_3_), 0.87 (d, *J =* 6.6 Hz, 6H, CH_3_-6, CH_3_-7), 0.67 (q, *J =* 12.4 Hz, 1H, CH_b_-3). ^13^C-NMR (DMSO-*d*6): δ 169.59, 152.40, 141.96, 138.84, 134.02, 129.22, 128.72 (2C), 125.51, 122.73, 120.44, 120.21, 119.96 (2C), 119.32, 60.48, 59.72 (2C), 45.00, 41.61, 33.79, 30.88 (2C), 19.32 (2C), 13.92. HRMS (ESI) *m*/*z*: [M + H]^+^ calculated for C_26_H_32_O_3_N_3_ClF_3_, 526.20788; found, 526.20862, Δ 1.41 ppm.

#### 3.1.13. Preparation of Racemic 3-[3-[3-(4-chlorophenyl)ureido]-4-(3,5-dimethylpiperidin-1-yl)-phenyl]-4,4,4-trifluorobutyric Acid (**i3**)

Hydrolysis of compound **g3** (134 mg, 0.26 mmol) following the general procedure B afforded compound **i3** (light yellow solid, 102 mg, 80.3% yield). mp: 123.5–124.7 °C. ^1^H-NMR (DMSO-*d*6): δ 12.51 (s, 1H, COOH), 9.70 (s, 1H, NH), 8.13 (d, *J =* 1.8 Hz, 1H, H-phenyl), 8.10 (s, 1H, NH), 7.55–7.49 (m, 2H, H-phenyl), 7.36–7.31 (m, 2H, H-phenyl), 7.14 (d, *J =* 8.2 Hz, 1H, H-phenyl), 6.99 (dd, *J =* 8.2, 1.9 Hz, 1H, H-phenyl), 3.95–3.83 (m, 1H, CH), 2.98–2.76 (m, 4H, CH_2_, CH_a_-1 and CH_a_-5), 2.15 (td, *J =* 11.2, 2.9 Hz, 2H, CH_b_-1 and CH_b_-5), 2.05–1.92 (m, 2H, CH-2, CH-4), 1.80 (d, *J =* 12.9 Hz, 1H, CH_a_-3), 0.87 (d, *J =* 6.6 Hz, 6H, CH_3_-6, CH_3_-7), 0.67 (q, *J =* 12.4 Hz, 1H, CH_b_-3). ^13^C-NMR (DMSO-*d*_6_): δ 171.08, 152.47, 141.93, 138.91, 134.13, 129.58, 128.73 (2C), 126.94, 125.52, 122.83, 120.46, 120.01 (2C), 119.34, 59.80 (2C), 45.20, 41.65, 33.86, 30.92 (2C), 19.32 (2C). HRMS (ESI) *m*/*z*: [M + H]^+^ calculated for C_24_H_28_O_3_N_3_ClF_3_, 498.17658; found, 498.17761, Δ 2.07 ppm.

#### 3.1.14. Preparation of Racemic 3-[3-(3-cyclohexylureido)-4-(3-methylpiperidin-1-yl)-phenyl]-4,4,4-trifluorobutyric Acid Ethyl Ester (**g4**)

Reaction of compound **f** and cyclohexyl isocyanate following the general procedure A afforded compound **g4** (white solid, 86.3% yield). ^1^H-NMR (DMSO-*d*_6_): δ 8.08 (s, 1H, NH), 7.63 (s, 1H, NH), 7.12–7.02 (m, 2H, H-phenyl), 6.89 (d, *J =* 8.2 Hz, 1H, H-phenyl), 4.06–3.93 (m, 2H, CH_2_), 3.92–3.80 (m, 1H, CH), 3.52–3.40 (m, 1H, CH-cyclohexyl), 3.04–2.75 (m, 4H, CH_2_, CH_a_-1 and CH_a_-5), 2.09 (dt, *J =* 11.0, 5.5 Hz, 2H, CH_b_-1 and CH_b_-5), 1.96 (d, *J =* 6.7 Hz, 2H, CH-2, CH-4), 1.87–1.49 (m, 6H, CH_a_-3, CH-cyclohexyl), 1.36–0.99 (m, 8H, CH-cyclohexyl, CH_3_), 0.86 (d, *J =* 6.5 Hz, 6H, CH_3_-6, CH_3_-7), 0.65 (q, *J =* 12.0 Hz, 1H, CH_b_-3). HRMS (ESI) *m*/*z*: [M + H]^+^ calculated for C_26_H_39_O_3_N_3_F_3_, 498.29380; found, 498.29480, Δ 2.00 ppm.

#### 3.1.15. Preparation of Racemic 3-[3-(3-cyclohexylureido)-4-(3,5-dimethylpiperidin-1-yl)phenyl]-4,4,4-trifluorobutyric Acid (**i4**)

Hydrolysis of compound **g4** (80 mg, 0.16 mmol) following the general procedure B afforded compound **i4** (white solid, 53 mg, 70.1% yield). mp: 134.2–135.7 °C. ^1^H-NMR (DMSO-*d*_6_): δ 7.91–7.76 (m, 1H, H-phenyl), 7.45–7.21 (m, 2H, NH, H-phenyl), 7.15–6.98 (m, 1H, H-phenyl), 4.00–3.85 (m, 1H, CH), 3.54–3.42 (m, 1H, CH-cyclohexyl), 3.14–2.75 (m, 4H, CH_a_-1 and CH_a_-5, CH_2_), 2.06–1.90 (m, 2H, CH_b_-1 and CH_b_-5), 1.85–1.76 (m, 3H, CH-2, CH-4, CH-cyclohexyl), 1.75–1.62 (m, 3H, CH_a_-3, CH-cyclohexyl), 1.60–1.48 (m, 1H, CH-cyclohexyl), 1.37–1.11 (m, 6H, CH-cyclohexyl), 0.89 (d, *J =* 6.6 Hz, 6H, CH_3_-6, CH_3_-7), 0.75 (q, *J =* 12.6 Hz, 1H, CH_b_-3). HRMS(ESI) *m*/*z*: [M + H]^+^ calculated for C_24_H_35_O_3_N_3_F_3_, 470.26250; found, 470.26343, Δ 1.97 ppm.

#### 3.1.16. Preparation of Racemic 3-[3-(3-butylureido)-4-(3,5-dimethylpiperidin-1-yl)-phenyl]-4,4,4-trifluorobutyric Acid Ethyl Ester (**g5**)

Reaction of compound **f** and butyl isocyanate following the general procedure A afforded compound **g5** (white solid, 84.1% yield). ^1^H-NMR (DMSO-*d*_6_): δ 8.08 (d, *J* = 1.2 Hz, 1H, H-phenyl), 7.65 (s, 1H, NH), 7.15 (t, *J* = 5.4 Hz, 1H, NH), 7.06 (d, *J* = 8.2 Hz, 1H, H-phenyl), 6.92–6.87 (m, 1H, H-phenyl), 4.06–3.93 (m, 2H, CH_2_), 3.93–3.81 (m, 1H, CH), 3.12–3.05 (m, 2H, CH-butyl), 3.04–2.76 (m, 4H, CH_2_, CH_a_-1 and CH_a_-5), 2.10 (td, *J* = 11.0, 2.9 Hz, 2H, CH_b_-1 and CH_b_-5), 2.03–1.88 (m, 2H, CH-2, CH-4), 1.78 (d, *J* = 12.7 Hz, 1H, CH_a_-3), 1.47–1.26 (m, 4H, CH-butyl), 1.07 (t, *J* = 7.1 Hz, 3H, CH_3_), 0.93–0.82 (m, 9H, CH- butyl, CH_3_-6, CH_3_-7), 0.65 (q, *J* = 12.0 Hz, 1H, CH_b_-3). HRMS (ESI) *m*/*z*: [M + H]^+^ calculated for C_24_H_37_O_3_N_3_F_3_, 472.27815; found, 472.27792, Δ −0.49 ppm.

#### 3.1.17. Preparation of Racemic 3-[3-(3-butylureido)-4-(3,5-dimethylpiperidin-1-yl)-phenyl]-4,4,4-trifluorobutyric Acid (**i5**)

Hydrolysis of compound **g5** (80 mg, 0.17 mmol) following the general procedure B afforded compound **i5** (white solid, 56 mg, 74.3% yield). mp: 131.0–132.9 °C. ^1^H-NMR (DMSO-*d*_6_): δ 8.00–7.80 (m, 1H, -phenyl), 7.36–7.15 (m, 2H, NH, H-phenyl), 7.09–6.96 (m, 1H, H-phenyl), 3.98–3.83 (m, 1H, H-phenyl), 3.11 (t, *J =* 6.6 Hz, 2H, CH- butyl), 3.04–2.76 (m, 4H, CH_2_, CH_a_-1 and CH_a_-5), 2.46–2.21 (m, 2H, CH_b_-1 and CH_b_-5), 2.05–1.89 (m, 2H, CH-2, CH-4), 1.80 (d, *J =* 12.6 Hz, 1H, CH_a_-3), 1.49–1.29 (m, 4H, CH-butyl), 0.95–0.83 (m, 9H, CH- butyl, CH_3_-6, CH_3_-7), 0.73 (q, *J =* 11.7 Hz, 1H, CH_b_-3). HRMS (ESI) *m*/*z*: [M + H]^+^ calculated for C_22_H_33_O_3_N_3_F_3_, 444.24685; found, 444.24750, Δ 1.46 ppm.

#### 3.1.18. Preparation of Racemic 3-[4-(3,5-dimethylpiperidin-1-yl)-3-(3-*o*-tolylureido)phenyl]-4,4,4-trifluorobutyric Acid Ethyl Ester (**g6**)

Reaction of compound **f** and *o*-tolyl isocyanate following the general procedure A afforded compound **g6** (white solid, 78.4% yield). ^1^H-NMR (DMSO-*d*_6_): δ 8.62 (s, 1H, NH), 8.07 (s, 1H, NH), 8.04–8.01 (m, 1H, H-phenyl) 7.58–7.54 (m, 1H, H-phenyl), 7.20–7.09 (m, 2H, H-phenyl), 7.06–6.90 (m, 3H, H-phenyl), 4.03–3.80 (m, 3H, CH_2_, CH), 3.01–2.75 (m, 4H, CH_2_, CH_a_-1 and CH_a_-5), 2.20 (s, 3H, CH_3_-phenyl), 2.05 (t, *J =* 10.4 Hz, 2H, CH_b_-1 and CH_b_-5), 1.84–1.66 (m, 3H, CH-2, CH-4, CH_a_-3), 1.03 (t, *J =* 7.0 Hz, 3H, CH_3_), 0.79 (d, *J =* 6.5 Hz, 6H, CH_3_-6, CH_3_-7), 0.57 (q, *J =* 12.0 Hz, 1H, CH_b_-3). HRMS (ESI) *m*/*z*: [M + H]^+^ calculated for C_27_H_35_O_3_N_3_F_3_, 506.26250; found, 506.26297, Δ 0.92 ppm.

#### 3.1.19. Preparation of Racemic 3-[4-(3,5-dimethylpiperidin-1-yl)-3-(3-*o*-tolylureido)phenyl]-4,4,4-trifluorobutyric Acid (**i6**)

Hydrolysis of compound **g6** (203 mg, 0.40 mmol) following the general procedure B afforded compound **i6** (white solid, 156 mg, 81.3% yield). mp: 109.3–111.2 °C. ^1^H-NMR (DMSO-*d*_6_): δ 12.53 (s, 1H, COOH), 8.67 (s, 1H, NH), 8.08 (s, 1H, NH), 8.04 (d, *J =* 1.0 Hz, 1H, H-phenyl), 7.56 (d, *J =* 7.7 Hz, 1H, H-phenyl), 7.25–7.14 (m, 2H, H-phenyl), 7.11–6.94 (m, 3H, H-phenyl), 3.96–3.80 (m, 1H, CH), 2.98–2.72 (m, 4H, CH_a_-1 and CH_a_-5, CH_2_), 2.25 (s, 3H, CH_3_-phenyl), 2.10 (td, *J =* 11.1, 3.7 Hz, 2H, CH_b_-1 and CH_b_-5), 1.86–1.68 (m, 3H, CH-2, CH-4, CH_a_-3), 0.84 (d, *J =* 6.4 Hz, 6H, CH_3_-6, CH_3_-7), 0.62 (q, *J =* 12.2 Hz, 1H, CH_b_-3). ^13^C-NMR (DMSO-*d*_6_): δ 171.12, 153.19, 142.24, 136.94, 133.95, 130.91, 130.47, 129.16, 126.93, 126.34, 124.53, 124.37, 122.80, 120.12, 119.96, 59.50 (2C), 45.13, 41.62, 33.87, 30.88 (2C), 19.31 (2C), 18.02. HRMS (ESI) *m*/*z*: [M + H]^+^ calculated for C_25_H_31_O_3_N_3_F_3_, 478.23120; found, 478.23215, Δ 1.98 ppm.

#### 3.1.20. Preparation of Racemic 3-[4-(3,5-dimethylpiperidin-1-yl)-3-(3-*m*-tolylureido)phenyl]-4,4,4-trifluorobutyric Acid Ethyl Ester (**g7**)

Reaction of compound **f** and *m*-tolyl isocyanate following the general procedure A afforded compound **g7** (white solid, 86.1% yield). ^1^H-NMR (DMSO-*d*_6_): δ 9.46 (s, 1H, NH), 8.14 (d, *J =* 1.9 Hz, 1H, H-phenyl), 8.05 (s, 1H, NH), 7.37–7.33 (m, 1H, H-phenyl), 7.28–7.23 (m, 1H, H-phenyl), 7.20–7.11 (m, 2H, H-phenyl), 6.98 (dd, *J =* 8.2, 1.9 Hz, 1H, H-phenyl), 6.81 (d, *J =* 7.4 Hz, 1H, H-phenyl), 4.07–3.87 (m, 3H, CH_2_, CH), 3.07–2.80 (m, 4H, CH_2_, CH_a_-1 and CH_a_-5), 2.29 (s, 3H, CH_3_-phenyl), 2.14 (td, *J =* 11.1, 3.0 Hz, 2H, CH_b_-1 and CH_b_-5), 2.04–1.89 (m, 2H, CH-2, CH-4), 1.80 (d, *J =* 13.2 Hz, 1H, CH_a_-3), 1.08 (t, *J =* 7.1, 3H, CH_3_), 0.87 (d, *J =* 6.6 Hz, 6H, CH_3_-6, CH_3_-7), 0.66 (q, *J =* 12.4 Hz, 1H, CH_b_-3). HRMS (ESI) *m*/*z*: [M + H]^+^ calculated for C_27_H_35_O_3_N_3_F_3_, 506.26250; found, 506.26297, Δ 0.92 ppm.

#### 3.1.21. Preparation of Racemic 3-[4-(3,5-dimethylpiperidin-1-yl)-3-(3-*m*-tolylureido)phenyl]-4,4,4-trifluorobutyric Acid (**i7**)

Hydrolysis of compound **g7** (209 mg, 0.41 mmol) following the general procedure B afforded compound **i7** (white solid, 148 mg, 75.0% yield). mp: 121.2–122.4 °C. ^1^H-NMR (DMSO-*d*_6_): δ 9.48 (s, 1H, NH), 8.14 (d, *J =* 1.5 Hz, 1H, H-phenyl), 8.06 (s, 1H, NH), 7.37–7.34 (m, 1H, H-phenyl), 7.26 (d, *J =* 8.6 Hz, 1H, H-phenyl), 7.20–7.10 (m, 2H, H-phenyl), 6.97 (dd, *J =* 8.2, 1.6 Hz, 1H, H-phenyl), 6.81 (d, *J =* 7.4 Hz, 1H, H-phenyl), 3.96–3.82 (m, 1H, CH), 3.00–2.74 (m, 4H, CH_2_, CH_a_-1 and CH_a_-5), 2.29 (s, 3H, CH_3_-phenyl), 2.14 (td, *J =* 11.0, 4.2 Hz, 2H, CH_b_-1 and CH_b_-5), 2.05–1.91 (m, 2H, CH-2, CH-4), 1.80 (d, *J =* 12.8 Hz, 1H, CH_a_-3), 0.87 (d, *J =* 6.6 Hz, 6H, CH_3_-6, CH_3_-7), 0.66 (q, *J =* 12.0 Hz, 1H, CH_b_-3). ^13^C-NMR (DMSO-*d*_6_): δ 171.16, 152.57, 141.84, 139.79, 138.09, 134.28, 129.56, 128.75, 126.94, 122.82, 122.60, 120.39, 119.30, 119.17, 115.78, 59.77 (2C), 45.21, 41.67, 33.93, 30.90 (2C), 21.30, 19.35 (2C). HRMS (ESI) *m*/*z*: [M + H]^+^ calculated for C_25_H_31_O_3_N_3_F_3_, 478.23120; found, 478.23227, Δ 2.23 ppm.

#### 3.1.22. Preparation of Racemic 3-[3-[3-(2-chlorophenyl)-ureido]-4-(3,5-dimethylpiperidin-1-yl)-phenyl]-4,4,4-trifluorobutyric Acid Ethyl Ester (**g8**)

Reaction of compound **f** and 2-chlorophenyl isocyanate following the general procedure A afforded compound **g8** (white solid, 81.7% yield). ^1^H-NMR (DMSO-*d*_6_): δ 9.03 (s, 1H, NH), 8.43 (s, 1H, NH), 7.97–7.90 (m, 2H, H-phenyl), 7.47 (dd, *J* = 8.0, 1.2 Hz, 1H, H-phenyl), 7.34–7.27 (m, 1H, H-phenyl), 7.13–6.99 (m, 3H, H-phenyl), 4.07–3.87 (m, 3H, CH_2_, CH), 3.07–2.82 (m, 4H, CH_2_, CH_a_-1 and CH_a_-5), 2.12 (t, *J* = 11.0 Hz, 2H, CH_b_-1 and CH_b_-5), 2.00–1.86 (m, 2H, CH-2, CH-4), 1.80 (d, *J* = 12.8 Hz, 1H, CH_a_-3), 1.08 (t, *J* = 7.2 Hz, 3H, CH_3_), 0.86 (d, *J* = 6.6 Hz, 6H, CH_3_-6, CH_3_-7), 0.65 (q, *J =* 12.0 Hz, 1H, CH_b_-3). HRMS (ESI) *m*/*z*: [M + H]^+^ calculated for C_26_H_32_O_3_N_3_ClF_3_, 526.20788; found, 526.20917, Δ 2.45 ppm.

#### 3.1.23. Preparation of Racemic 3-[3-[3-(2-chlorophenyl)ureido]-4-(3,5-dimethylpiperidin-1-yl)-phenyl]-4,4,4-trifluorobutyric Acid (**i8**)

Hydrolysis of compound **g8** (60 mg, 0.11 mmol) following the general procedure B afforded compound **i8** (white solid, 46 mg, 80.7% yield). mp: 111.2–113.7 °C. ^1^H-NMR (DMSO-*d*_6_): δ 12.43 (s, 1H, COOH), 9.01 (s, 1H, NH), 8.40 (s, 1H, NH), 7.95–7.85 (m, 2H, H-phenyl), 7.43 (d, *J =* 7.9 Hz, 1H, H-phenyl), 7.31–7.22 (m, 1H, H-phenyl), 7.09–6.93 (m, 3H, H-phenyl), 3.89–3.79 (m, 1H, CH), 2.93–2.71 (m, 4H, CH_2_, CH_a_-1 and CH_a_-5), 2.08 (td, *J =* 11.0, 2.6 Hz, 2H, CH_b_-1 and CH_b_-5), 1.96–1.87 (m, 2H, CH-2, CH-4), 1.74 (d, *J =* 12.7 Hz, 1H, CH_a_-3), 0.82 (d, *J =* 6.5 Hz, 6H, CH_3_-6, CH_3_-7), 0.61 (q, *J =* 12.0 Hz, 1H, CH_b_-3). HRMS (ESI) *m*/*z*: [M + H]^+^ calculated for C_24_H_28_O_3_N_3_ClF_3_, 498.17658; found, 498.17786, Δ 2.57 ppm.

#### 3.1.24. Preparation of Racemic 3-[3-[3-(3-chlorophenyl)ureido]-4-(3,5-dimethylpiperidin-1-yl)-phenyl]-4,4,4-trifluorobutyric Acid Ethyl Ester (**g9**)

Reaction of compound **f** and 3-chlorophenyl isocyanate following the general procedure A afforded compound **g9** (white solid, 79.7% yield). ^1^H-NMR (DMSO-*d*_6_): δ 9.76 (s, 1H, NH), 8.15–8.09 (m, 2H, H-phenyl, NH), 7.79–7.75 (m, 1H, H-phenyl), 7.35–7.25 (m, 2H, H-phenyl), 7.15 (d, *J =* 8.2 Hz, 1H, H-phenyl), 7.05–6.98 (m, 2H, H-phenyl), 4.06–3.92 (m, 3H, CH_2_, CH), 3.08–2.79 (m, 4H, CH_2_, CH_a_-1 and CH_a_-5), 2.15 (t, *J =* 11.0 Hz, 2H, CH_b_-1 and CH_b_-5), 2.05–1.90 (m, 2H, CH-2, CH-4), 1.78 (d, *J =* 12.0 Hz, 1H, CH_a_-3), 1.08 (t, *J =* 7.1 Hz, 3H, CH_3_), 0.87 (d, *J =* 6.5 Hz, 6H, CH_3_-6, CH_3_-7), 0.67 (q, *J =* 12.4 Hz, 1H, CH_b_-3). HRMS (ESI) *m*/*z*: [M + H]^+^ calculated for C_26_H_32_O_3_N_3_ClF_3_, 526.20788; found, 526.20776, Δ −0.23 ppm.

#### 3.1.25. Preparation of Racemic 3-[3-[3-(3-chlorophenyl)ureido]-4-(3,5-dimethylpiperidin-1-yl)-phenyl]-4,4,4-trifluorobutyric Acid (**i9**)

Hydrolysis of compound **g9** (300 mg, 0.57 mmol) following the general procedure B afforded compound **i9** (white solid, 234 mg, 82.5% yield). mp: 129.3–131.7 °C. ^1^H-NMR (DMSO-*d*_6_) δ 12.50 (s, 1H, COOH), 9.77 (s, 1H, NH), 8.14–8.09 (m, 2H, NH, H-phenyl), 7.77 (t, *J =* 1.9 Hz, 1H, H-phenyl), 7.35–7.25 (m, 2H, H-phenyl), 7.15 (d, *J =* 8.1 Hz, 1H, H-phenyl), 7.07–6.97 (m, 2H, H-phenyl), 3.98–3.84 (m, 1H, CH), 3.01–2.75 (m, 4H, CH_2_, CH_a_-1 and CH_a_-5), 2.15 (td, *J =* 11.1, 2.9 Hz, 2H, CH_b_-1 and CH_b_-5), 2.06–1.91 (m, 2H, CH-2, CH-4), 1.81 (d, *J =* 12.9 Hz, 1H, CH_a_-3), 0.87 (d, *J =* 6.6 Hz, 6H, CH_3_-6, CH_3_-7), 0.67 (q, *J =* 12.1 Hz, 1H, CH_b_-3). ^13^C-NMR (DMSO-*d*_6_): δ 171.03, 152.36, 141.97, 141.42, 133.92, 133.33, 130.49, 129.55, 126.86, 122.91, 121.60, 120.48, 119.38, 117.78, 116.74, 59.75 (2C), 45.10, 41.61, 33.78, 30.90 (2C), 19.32 (2C). HRMS (ESI) *m*/*z*: [M + H]^+^ calculated for C_24_H_28_O_3_N_3_ClF_3_, 498.17658; found, 498.17770, Δ 2.25 ppm.

#### 3.1.26. Preparation of Racemic 3-[3-[3-(2-cyanophenyl)ureido]-4-(3,5-dimethylpiperidin-1-yl)-phenyl]-4,4,4-trifluorobutyric Acid Ethyl Ester (**g10**)

Reaction of compound **f** and 2-cyanophenyl isocyanate following the general procedure A afforded compound **g10** (white solid, 79.1% yield). ^1^H-NMR (DMSO-*d*_6_): δ 10.17 (s, 1H, NH), 8.28 (s, 1H, NH), 8.17–8.11 (m, 2H, H-phenyl), 8.03 (dd, *J =* 8.0, 3.8 Hz, 1H, H-phenyl), 7.63–7.57 (m, 1H, H-phenyl), 7.22–7.01 (m, 3H, H-phenyl), 4.07–3.87 (m, 3H, CH_2_, CH), 3.08–2.82 (m, 4H, CH_2_, CH_a_-1 and CH_a_-5), 2.13 (t, *J* = 11.0 Hz, 2H, CH_b_-1 and CH_b_-5), 2.03–1.88 (m, 2H, CH-2, CH-4), 1.80 (d, *J =* 12.8 Hz, 1H, CH_a_-3), 1.07 (t, *J =* 7.1 Hz, 3H, CH_3_), 0.86 (d, *J =* 6.6 Hz, 6H, CH_3_-6, CH_3_-7), 0.65 (q, *J* = 12.4 Hz, 1H, CH_b_-3). HRMS (ESI) *m*/*z*: [M + H]^+^ calculated for C_27_H_32_O_3_N_4_F_3_, 517.24210; found, 517.24335, Δ 2.41 ppm.

#### 3.1.27. Preparation of Racemic 3-[3-[3-(2-cyanophenyl)ureido]-4-(3,5-dimethylpiperidin-1-yl)-phenyl]-4,4,4-trifluorobutyric Acid (**i10**)

Hydrolysis of compound **g10** (40 mg, 0.08 mmol) following the general procedure B afforded compound **i10** (white solid, 30 mg, 79.8% yield). mp: 178.2–179.9 °C. ^1^H-NMR (DMSO-*d*_6_): δ 12.51 (s, 1H, COOH), 10.18 (s, 1H, NH), 8.28 (s, 1H, NH), 8.16–8.10 (m, 2H, H-phenyl), 8.02 (dd, *J =* 8.1, 3.6 Hz, 1H, H-phenyl), 7.62–7.56 (m, 1H, H-phenyl), 7.22–7.01 (m, 3H, H-phenyl), 3.99–3.84 (m, 1H, CH), 3.02–2.76 (m, 4H, CH_2_, CH_a_-1 and CH_a_-5), 2.14 (t, *J* = 10.6 Hz, 2H, CH_b_-1 and CH_b_-5), 2.04–1.89 (m, 2H, CH-2, CH-4), 1.80 (d, *J =* 12.8 Hz, 1H, CH_a_-3), 0.87 (d, *J =* 6.6 Hz, 6H, CH_3_-6, CH_3_-7), 0.66 (q, *J* = 12.4 Hz, 1H, CH_b_-3). HRMS (ESI) *m*/*z*: [M + H]^+^ calculated for C_25_H_28_O_3_N_4_F_3_, 489.20987; found, 489.21080, Δ −1.90 ppm.

#### 3.1.28. Preparation of Racemic 3-[3-[3-(3-cyanophenyl)ureido]-4-(3,5-dimethylpiperidin-1-yl)-phenyl]-4,4,4-trifluorobutyric Acid Ethyl Ester (**g11**)

Reaction of compound **f** and 3-cyanophenyl isocyanate following the general procedure A afforded compound **g11** (white solid, 87.5% yield). ^1^H-NMR (DMSO-*d*_6_): δ 9.91 (s, 1H, NH), 8.17 (s, 1H, NH), 8.14–8.11 (m, 1H, H-phenyl), 8.06–8.03 (m, 1H, H-phenyl), 7.66 (d, *J =* 8.2 Hz, 1H, H-phenyl), 7.54–7.41 (m, 2H, H-phenyl), 7.16 (d, *J =* 8.1 Hz, 1H, H-phenyl), 7.02 (d, *J =* 8.4 Hz, 1H, H-phenyl), 4.08–3.88 (m, 3H, CH_2_, CH), 3.07–2.81 (m, 4H, CH_2_, CH_a_-1 and CH_a_-5), 2.15 (t, *J =* 10.8 Hz, 2H, CH_b_-1 and CH_b_-5), 2.06–1.92 (m, 2H, CH-2, CH-4), 1.81 (d, *J =* 12.8 Hz, 1H, CH_a_-3), 1.08 (t, *J =* 7.2 Hz, 3H, CH_3_), 0.87 (d, *J =* 6.5 Hz, 6H, CH_3_-6, CH_3_-7), 0.67 (q, *J =* 12.1 Hz, 1H, CH_b_-3). HRMS (ESI) *m*/*z*: [M + H]^+^ calculated for C_27_H_32_O_3_N_4_F_3_, 517.24210; found, 517.24121, Δ −1.72 ppm.

#### 3.1.29. Preparation of Racemic 3-[3-[3-(3-cyanophenyl)ureido]-4-(3,5-dimethylpiperidin-1-yl)-phenyl]-4,4,4-trifluorobutyric Acid (**i11**)

Hydrolysis of compound **g11** (180 mg, 0.35 mmol) following the general procedure B afforded compound **i11** (white solid, 121 mg, 71.2% yield). mp: 128.4–130.2 °C. ^1^H-NMR (DMSO-*d*_6_): δ 12.52 (s, 1H, COOH), 9.94 (s, 1H, NH), 8.01–8.23 (m, 3H, NH, H-phenyl), 7.66 (d, *J* = 7.3 Hz, 1H, H-phenyl), 7.57–7.39 (m, 2H, H-phenyl), 7.16 (d, *J* = 8.2 Hz, 1H, H-phenyl), 7.01 (d, *J* = 8.2 Hz, 1H, H-phenyl), 3.98–3.83 (m, 1H, CH), 3.04–2.74 (m, 4H, CH_2_, CH_a_-1 and CH_a_-5), 2.16 (t, *J =* 9.9 Hz, 2H, CH_b_-1 and CH_b_-5), 2.06–1.92 (m, 2H, CH-2, CH-4), 1.81 (d, *J* = 11.3 Hz, 1H, CH_a_-3), 0.87 (d, *J* = 6.1 Hz, 6H, CH_3_-6, CH_3_-7), 0.67 (q, *J* = 11.6 Hz, 1H, CH_b_-3). ^13^C-NMR (DMSO-*d*_6_): δ 171.03, 152.38, 142.03, 140.76, 133.79, 130.31, 129.58, 126.90, 125.47, 123.06, 122.96, 120.92, 120.52, 119.40, 118.91, 111.72, 59.75 (2C), 45.10, 41.60, 33.78, 30.90 (2C), 19.33 (2C). HRMS (ESI) *m*/*z*: [M + H]^+^ calculated for C_25_H_28_O_3_N_4_F_3_, 489.21080; found, 489.21274, Δ 1.94 ppm.

#### 3.1.30. Preparation of Racemic 3-[3-[3-(4-cyanophenyl)ureido]-4-(3,5-dimethylpiperidin-1-yl)-phenyl]-4,4,4-trifluorobutyric Acid Ethyl Ester (**g12**)

Reaction of compound **f** and 4-cyanophenyl isocyanate following the general procedure A afforded compound **g12** (white solid, 84.1% yield). ^1^H-NMR (DMSO-*d_6_*): δ 10.05 (s, 1H, NH), 8.21 (s, 1H, NH), 8.12 (d, *J =* 1.8 Hz, 1H, H-phenyl), 7.77–7.72 (m, 2H, H-phenyl), 7.70–7.66 (m, 2H, H-phenyl), 7.16 (d, *J* = 8.3 Hz, 1H, H-phenyl), 7.03 (dd, *J* = 8.1, 2.0 Hz, 1H, H-phenyl), 4.07–3.88 (m, 3H, CH_2_, CH), 3.08–2.82 (m, 4H, CH_2_, CH_a_-1 and CH_a_-5), 2.15 (t, *J* = 10.9 Hz, 2H, CH_b_-1 and CH_b_-5), 2.05–1.90 (m, 2H, CH-2, CH-4), 1.81 (d, *J* = 12.7 Hz, 1H, CH_a_-3), 1.07 (t, *J =* 7.1 Hz, 3H, CH_3_), 0.90–0.81 (m, 6H, CH_3_-6, CH_3_-7), 0.67 (q, *J* = 12.8 Hz, 1H, CH_b_-3). HRMS (ESI) *m*/*z*: [M + H]^+^ calculated for C_27_H_32_O_3_N_4_F_3_, 517.24210; found, 517.24298, Δ 1.70 ppm.

#### 3.1.31. Preparation of Racemic 3-[3-[3-(4-cyanophenyl)ureido]-4-(3,5-dimethylpiperidin-1-yl)-phenyl]-4,4,4-trifluorobutyric Acid (**i12**)

Hydrolysis of compound **g12** (331 mg, 0.64 mmol) following the general procedure B afforded compound **i12** (white solid, 271 mg, 86.5% yield). mp: 172.3–172.9 °C. ^1^H-NMR (DMSO-*d_6_*): δ 10.18 (s, 1H, NH), 8.27 (s, 1H, NH), 8.12 (d, *J =* 1.4 Hz, 1H, H-phenyl), 7.76–7.67 (m, 4H, H-phenyl), 7.16 (dd, *J =* 8.0, 4.2 Hz, 1H, H-phenyl), 7.02 (dd, *J =* 8.2, 1.8 Hz, 1H, H-phenyl), 3.98–3.83 (m, 1H, CH), 3.03–2.75 (m, 4H, CH_2_, CH_a_-1 and CH_a_-5), 2.16 (td, *J =* 11.1, 1.9 Hz, 2H, CH_b_-1 and CH_b_-5), 2.06–1.94 (m, 2H, CH-2, CH-4), 1.80 (d, *J =* 12.7 Hz, 1H, CH_a_-3), 0.86 (d, *J =* 6.5 Hz, 6H, CH_3_-6, CH_3_-7), 0.67 (q, *J* = 12.4 Hz, 1H, CH_b_-3). ^13^C-NMR (DMSO-*d_6_*): δ 170.92, 152.17, 144.35, 141.88, 133.53, 133.25 (2C), 129.64, 126.77 (q, J_c-f_ = 280.8 Hz), 123.29, 120.49, 119.75, 119.31, 118.11 (2C), 103.25, 59.73 (2C), 44.97 (q, *J* = 27.3 Hz), 41.45, 33.67, 30.62 (2C), 19.21 (2C). HRMS (ESI) *m*/*z*: [M + H]^+^ calculated for C_25_H_28_O_3_N_4_F_3_, 489.21080; found, 489.20990, Δ −1.84 ppm.

#### 3.1.32. Preparation of Racemic 3-{4-(3,5-dimethylpiperidin-1-yl)-3-[3-(2-methoxyphenyl)ureido]-phenyl}-4,4,4-trifluorobutyric Acid Ethyl Ester (**g13**)

Reaction of compound **f** and 2-methoxyphenyl isocyanate following the general procedure A afforded compound **g13** (white solid, 85.9% yield). ^1^H-NMR (DMSO-*d_6_*): δ 8.91 (s, 1H, NH), 8.42 (s, 1H, NH), 8.00–7.93 (m, 2H, H-phenyl), 7.10–6.96 (m, 4H, H-phenyl), 6.93–6.86 (m, 1H, H-phenyl), 4.07–3.87 (m, 3H, CH_2_, CH), 3.85 (s, 3H, OCH_3_-phenyl), 3.05–2.83 (m, 4H, CH_2_, CH_a_-1 and CH_a_-5), 2.10 (td, *J =* 11.1, 2.1 Hz, 2H, CH_b_-1 and CH_b_-5), 2.02–1.87 (m, 2H, CH-2, CH-4), 1.77 (d, *J =* 12.8 Hz, 1H, CH_a_-3), 1.07 (t, *J =* 7.1 Hz, 3H, CH_3_), 0.85 (d, *J =* 6.6 Hz, 6H, CH_3_-6, CH_3_-7), 0.64 (q, *J* = 12.4 Hz, 1H, CH_b_-3). HRMS (ESI) *m*/*z*: [M + H]^+^ calculated for C_27_H_35_O_4_N_3_F_3_, 522.25742; found, 522.25598, Δ −2.75 ppm.

#### 3.1.33. Preparation of Racemic 3-{4-(3,5-dimethylpiperidin-1-yl)-3-[3-(2-methoxyphenyl)ureido]-phenyl}-4,4,4-trifluorobutyric Acid (**i13**)

Hydrolysis of compound **g13** (80 mg, 0.15 mmol) following the general procedure B afforded compound **i13** (white solid, 61 mg, 80.8% yield). mp: 119.3–120.2 °C. ^1^H-NMR (DMSO-*d*_6_): δ 12.64 (s, 1H, COOH), 8.91 (s, 1H, NH), 8.42 (s, 1H, NH), 7.99–7.93 (m, 2H, H-phenyl), 7.10–6.86 (m, 5H, H-phenyl), 3.94–3.81 (m, 4H, CH, OCH_3_-phenyl), 2.93–2.65 (m, 4H, CH_a_-1 and CH_a_-5, CH_2_), 2.10 (td, *J* = 11.2, 4.6 Hz, 2H, CH_b_-1 and CH_b_-5), 2.01–1.86 (m, 2H, CH-2, CH-4), 1.77 (d, *J* = 13.3 Hz, 1H, CH_a_-3), 0.85 (d, *J* = 6.5 Hz, 6H, CH_3_-6, CH_3_-7), 0.63 (q, *J* = 12.4 Hz, 1H, CH_b_-3). HRMS (ESI) *m*/*z*: [M + H]^+^ calculated for C_25_H_31_O_4_N_3_F_3_, 494.22612; found, 494.22455, Δ −3.17 ppm.

#### 3.1.34. Preparation of Racemic 3-{4-(3,5-dimethylpiperidin-1-yl)-3-[3-(3-methoxyphenyl)ureido]-phenyl}-4,4,4-trifluorobutyric Acid Ethyl Ester (**g14**)

Reaction of compound **f** and 3-methoxyphenyl isocyanate following the general procedure A afforded compound **g14** (white solid, 83.9% yield). ^1^H-NMR (DMSO-*d*_6_): δ 9.55 (s, 1H, NH), 8.13 (d, *J =* 1.9 Hz, 1H, H-phenyl), 8.07 (s, 1H, NH), 7.22–7.17 (m, 2H, H-phenyl), 7.13 (d, *J =* 8.2 Hz, 1H, H-phenyl), 7.03–6.96 (m, 2H, H-phenyl), 6.59–6.55 (m, 1H, H-phenyl), 4.07–3.88 (m, 3H, CH_2_, CH), 3.75 (s, 3H, OCH_3_-phenyl), 3.06–2.75 (m, 4H, CH_2_, CH_a_-1 and CH_a_-5), 2.13 (td, *J =* 11.2, 1.6 Hz, 2H, CH_b_-1 and CH_b_-5), 2.04–1.91 (m, 2H, CH-2, CH-4), 1.80 (d, *J =* 12.7 Hz, 1H, CH_a_-3), 1.10-1.06 (t, *J =* 7.2 Hz, 3H, CH_3_), 0.87 (d, *J =* 6.6 Hz, 6H, CH_3_-6, CH_3_-7), 0.66 (q, *J* = 12.4 Hz, 1H, CH_b_-3). HRMS (ESI) *m*/*z*: [M + H]^+^ calculated for C_27_H_35_O_4_N_3_F_3_, 522.25742; found, 522.25769, Δ 0.52 ppm.

#### 3.1.35. Preparation of Racemic 3-{4-(3,5-dimethylpiperidin-1-yl)-3-[3-(3-methoxyphenyl)ureido]-phenyl}-4,4,4-trifluorobutyric Acid (**i14**)

Hydrolysis of compound **g14** (245 mg, 0.47 mmol) following the general procedure B afforded compound **i14** (white solid, 178 mg, 76.7% yield). mp: 112.3–113.7 °C. ^1^H-NMR (DMSO-*d*_6_): δ 12.52 (s, 1H, COOH), 9.56 (s, 1H, NH), 8.14 (d, *J =* 1.6 Hz, 1H, H-phenyl), 8.08 (s, 1H, NH), 7.22–7.11 (m, 3H, H-phenyl), 7.03–6.96 (m, 2H, H-phenyl), 6.57 (dd, *J =* 8.2, 2.5 Hz, 1H, H-phenyl), 3.96–3.81 (m, 1H, CH), 3.75 (s, 3H, OCH_3_-phenyl), 3.00–2.77 (m, 4H, CH_2_, CH_a_-1 and CH_a_-5), 2.14 (td, *J =* 11.0, 2.1 Hz, 2H, CH_b_-1 and CH_b_-5), 2.05–1.92 (m, 2H, CH-2, CH-4), 1.80 (d, *J =* 12.9 Hz, 1H, CH_a_-3), 0.87 (d, *J =* 6.6 Hz, 6H, CH_3_-6, CH_3_-7), 0.67 (q, *J =* 12.0 Hz, 1H, CH_b_-3). ^13^C-NMR (DMSO-*d*_6_): δ 171.08, 159.82, 152.50, 141.89, 141.09, 134.19, 129.66, 129.52, 126.90, 122.58, 120.44, 119.38, 110.87, 107.65, 104.23, 59.74 (2C), 55.06, 45.15, 41.64, 33.83, 30.88 (2C), 19.34 (2C). HRMS (ESI) *m*/*z*: [M + H]^+^ calculated for C_25_H_31_O_4_N_3_F_3_, 494.22612; found, 494.22699, Δ 1.77 ppm.

#### 3.1.36. Preparation of Racemic 3-{4-(3,5-dimethylpiperidin-1-yl)-3-[3-(4-methoxyphenyl)ureido]-phenyl}-4,4,4-trifluorobutyric Acid Ethyl Ester (**g15**)

Reaction of compound **f** and 4-methoxyphenyl isocyanate following the general procedure A afforded compound **g15** (light yellow solid, 81.3% yield). ^1^H-NMR (DMSO-*d*_6_): δ 9.34 (s, 1H, NH), 8.16 (d, *J =* 1.9 Hz, 1H, H-phenyl), 7.99 (s, 1H, NH), 7.41–7.36 (m, 2H, H-phenyl), 7.12 (d, *J =* 8.2 Hz, 1H, H-phenyl), 6.96 (dd, *J =* 8.2, 1.9 Hz, 1H, H-phenyl), 6.91–6.87 (m, 2H, H-phenyl), 4.05–3.88 (m, 3H, CH_2_, CH), 3.73 (s, 3H, OCH_3_-phenyl), 3.06–2.80 (m, 4H, CH_2_, CH_a_-1 and CH_a_-5), 2.12 (dt, *J =* 11.2, 5.6 Hz, 2H, CH_b_-1 and CH_b_-5), 1.99–1.86 (m, 2H, CH-2, CH-4), 1.78 (d, *J =* 13.0 Hz, 1H, CH_a_-3), 1.08 (t, *J =* 7.2 Hz, 3H, CH_3_), 0.86 (d, *J =* 6.8 Hz, 6H, CH_3_-6, CH_3_-7), 0.66 (q, *J =* 12.0 Hz, 1H, CH_b_-3). HRMS (ESI) *m*/*z*: [M + H]^+^ calculated for C_27_H_35_O_4_N_3_F_3_, 522.25665; found, 522.25742, Δ −1.47 ppm.

#### 3.1.37. Preparation of Racemic 3-{4-(3,5-dimethylpiperidin-1-yl)-3-[3-(4-methoxyphenyl)ureido]-phenyl}-4,4,4-trifluorobutyric Acid (**i15**)

Hydrolysis of compound **g15** (230 mg, 0.44 mmol) following the general procedure B afforded compound **i15** (white solid, 181 mg, 81.8% yield). mp: 108.9–120.1 °C. ^1^H-NMR (DMSO-*d*_6_): δ 12.50 (s, 1H, COOH), 9.32 (s, 1H, NH), 8.15 (d, *J =* 1.9 Hz, 1H, H-phenyl), 7.99 (s, 1H, NH), 7.40–7.36 (m, 2H, H-phenyl), 7.12 (d, *J =* 8.3, 1H, H-phenyl), 6.95 (dd, *J =* 8.2, 1.9 Hz, 1H, H-phenyl), 6.90–6.86 (m, 2H, H-phenyl), 3.94–3.83 (m, 1H, CH), 3.73 (s, 3H, OCH_3_-phenyl), 2.98–2.75 (m, 4H, CH_2_, CH_a_-1 and CH_a_-5), 2.13 (td, *J =* 11.1, 3.1 Hz, 2H, CH_b_-1 and CH_b_-5), 1.98-1.85 (d, *J =* 6.6 Hz, 2H, CH-2, CH-4), 1.78 (d, *J =* 12.8 Hz, 1H, CH_a_-3), 0.86 (d, *J =* 6.6 Hz, 6H, CH_3_-6, CH_3_-7), 0.66 (q, *J =* 12.4 Hz, 1H, CH_b_-3). HRMS (ESI) *m*/*z*: [M + H]^+^ calculated for C_25_H_31_O_4_N_3_F_3_, 494.22612; found, 494.22815, Δ 4.11 ppm.

#### 3.1.38. Preparation of Racemic 3-[3-[3-(2,4-difluorophenyl)ureido]-4-(3,5-dimethylpiperidin-1-yl)-phenyl]-4,4,4-trifluorobutyric Acid Ethyl Ester (**g16**)

Reaction of compound **f** and 2,4-difluorophenyl isocyanate following the general procedure A afforded compound **g16** (white solid, 80.8% yield). ^1^H-NMR (DMSO-*d*_6_): δ 9.36 (s, 1H, NH), 8.34 (s, 1H, NH), 8.09–7.98 (m, 2H, H-phenyl), 7.31 (ddd, *J =* 11.6, 8.9, 2.9 Hz, 1H, H-phenyl), 7.14–6.97 (m, 3H, H-phenyl), 4.07–3.85 (m, 3H, CH_2_, CH), 3.06–2.82 (m, 4H, CH_2_, CH_a_-1 and CH_a_-5), 2.13 (t, *J =* 10.8 Hz, 2H, CH_b_-1 and CH_b_-5), 2.04–1.89 (m, 2H, CH-2, CH-4), 1.79 (d, *J =* 13.0 Hz, 1H, CH_a_-3), 1.07 (t, *J =* 7.1 Hz, 3H, CH_3_), 0.86 (d, *J =* 6.6 Hz, 6H, CH_3_-6, CH_3_-7), 0.66 (q, *J =* 12.4 Hz, 1H, CH_b_-3). HRMS (ESI) *m*/*z*: [M + H]^+^ calculated for C_26_H_31_O_3_N_3_F_5_, 528.22801; found, 528.22894, Δ 1.76 ppm.

#### 3.1.39. Preparation of Racemic 3-[3-[3-(2,4-difluorophenyl)ureido]-4-(3,5-dimethylpiperidin-1-yl)-phenyl]-4,4,4-trifluorobutyric Acid (**i16**)

Hydrolysis of compound **g16** (190 mg, 0.36 mmol) following the general procedure B afforded compound **i16** (white solid, 149 mg, 82.0% yield). mp: 119.6–120.6 °C. ^1^H-NMR (DMSO-*d*_6_): δ 9.38 (s, 1H, -NH-), 8.36 (s, 1H, -NH-), 8.08–7.97 (m, 2H, H-phenyl), 7.31 (ddd, *J =* 11.6, 8.9, 2.9 Hz, 1H, H-phenyl), 7.14–6.95 (m, 3H, H-phenyl), 4.07–3.86 (m, 1H, CH), 2.94–2.70 (m, 4H, CH_2_, CH_a_-1 and CH_a_-5), 2.13 (td, *J =* 11.0, 3.8 Hz, 2H, CH_b_-1 and CH_b_-5), 2.04–1.88 (m, 2H, CH-2, CH-4), 1.79 (d, *J =* 12.6 Hz, 1H, CH_a_-3), 0.86 (d, *J =* 6.5 Hz, 6H, CH_3_-6, CH_3_-7), 0.65 (q, *J =* 12.0 Hz, 1H, CH_b_-3). ^13^C-NMR (DMSO-*d*_6_): δ 171.31, 155.16 (2C), 152.71, 142.28, 133.78, 129.47, 126.97, 123.85, 123.82, 123.10, 120.20, 120.08, 111.07, 103.92, 59.65 (2C), 45.23, 41.65, 34.18, 30.77 (2C), 19.33 (2C). HRMS (ESI) *m*/*z*: [M + H]^+^ calculated for C_24_H_27_O_3_N_3_F_5_, 500.19671; found, 500.19440, Δ −4.62 ppm.

#### 3.1.40. Preparation of Racemic 3-[3-[3-(2,4-dichlorophenyl)ureido]-4-(3,5-dimethylpiperidin-1-yl)-phenyl]-4,4,4-trifluorobutyric Acid Ethyl Ester (**g17**)

Reaction of compound **f** and 2,4-dichlorophenyl isocyanate following the general procedure A afforded compound **g17** (white solid, 81.3% yield). ^1^H-NMR (DMSO-*d*_6_): δ 9.12 (s, 1H, NH), 8.50 (s, 1H, NH), 8.00 (d, *J =* 8.9 Hz, 1H, H-phenyl), 7.93 (d, *J =* 1.8 Hz, 1H, H-phenyl), 7.63 (d, *J =* 2.5 Hz, 1H, H-phenyl), 7.39 (dd, *J =* 8.9, 2.5 Hz, 1H, H-phenyl), 7.10 (d, *J =* 8.3 Hz, 1H, H-phenyl), 7.03 (dd, *J =* 8.2, 1.9 Hz, 1H, H-phenyl), 4.07–3.86 (m, 3H, CH_2_, CH), 3.06–2.82 (m, 4H, CH_2_, CH_a_-1 and CH_a_-5), 2.12 (t, *J =* 11.1 Hz, 2H, CH_b_-1 and CH_b_-5), 2.02–1.88 (m, 2H, CH-2, CH-4), 1.79 (d, *J =* 12.7 Hz, 1H, CH_a_-3), 1.07 (t, *J =* 7.1 Hz, 3H, CH_3_), 0.86 (d, *J =* 6.6 Hz, 6H, CH_3_-6, CH_3_-7), 0.65 (q, *J =* 12.4 Hz, 1H, CH_b_-3). HRMS (ESI) *m*/*z*: [M + H]^+^ calculated for C_26_H_31_O_3_N_3_Cl_2_F_3_, 560.16891; found, 560.17065, Δ 3.11 ppm.

#### 3.1.41. Preparation of Racemic 3-[3-[3-(2,4-dichlorophenyl)ureido]-4-(3,5-dimethylpiperidin-1-yl)-phenyl]-4,4,4-trifluorobutyric Acid (**i17**)

Hydrolysis of compound **g17** (80 mg, 0.14 mmol) following the general procedure B afforded compound **i17** (white solid, 61 mg, 80.1% yield). mp: 119.3–110.5 °C. ^1^H-NMR (DMSO-*d*_6_): δ 9.14 (s, 1H, NH), 8.50 (s, 1H, NH), 8.00 (d, *J =* 8.9 Hz, 1H, H-phenyl), 7.92 (d, *J =* 1.7 Hz, 1H, H-phenyl), 7.63 (d, *J =* 2.4 Hz, 1H, H-phenyl), 7.38 (dd, *J =* 8.9, 2.5 Hz, 1H, H-phenyl), 7.09 (d, *J =* 8.3 Hz, 1H, H-phenyl), 7.01 (dd, *J =* 8.2, 1.8 Hz, 1H, H-phenyl), 3.96–3.82 (m, 1H, CH), 2.96–1.71 (m, 4H, CH_a_-1 and CH_a_-5, CH_2_), 2.12 (td, *J =* 11.1, 3.9 Hz, 2H, CH_b_-1 and CH_b_-5), 2.03–1.87 (m, 2H, CH-2, CH-4), 1.83–1.75 (m, 1H, CH_a_-3), 0.86 (d, *J =* 6.6 Hz, 6H, CH_3_-6, CH_3_-7), 0.65 (q, *J =* 12.4 Hz, 1H, CH_b_-3). ^13^C-NMR (DMSO-*d*_6_): δ 171.29, 152.61, 143.00, 135.17, 133.12, 129.07, 128.72, 127.56, 127.24, 126.97, 125.07, 124.91, 123.65, 121.31, 119.86, 59.44 (2C), 45.12, 41.66, 34.15, 30.75 (2C), 19.33 (2C). HRMS (ESI) *m*/*z*: [M + H]^+^ calculated for C_24_H_27_O_3_N_3_Cl_2_F_3_, 532.13761; found, 532.13892, Δ 2.47 ppm.

#### 3.1.42. Preparation of Racemic 3-{4-(3,5-dimethylpiperidin-1-yl)-3-[3-(4-fluorophenyl)ureido]-phenyl}-4,4,4-trifluorobutyric Acid Ethyl Ester (**g18**)

Reaction of compound **f** and 4-fluorophenyl isocyanate following the general procedure A afforded compound **g18** (white solid, 81.3% yield). ^1^H-NMR (DMSO-*d*_6_): δ 9.57 (s, 1H, NH), 8.13 (d, *J =* 1.8 Hz, 1H, H-phenyl), 8.04 (s, 1H, NH), 7.54–7.46 (m, 2H, H-phenyl), 7.18–7.09 (m, 3H, H-phenyl), 6.98 (dd, *J =* 8.2, 1.9 Hz, 1H, H-phenyl), 4.08–3.85 (m, 3H, CH_2_, CH), 3.07–2.79 (m, 4H, CH_2_, CH_a_-1 and CH_a_-5), 2.14 (t, *J =* 10.6 Hz, 2H, CH_b_-1 and CH_b_-5), 2.05–1.88 (m, 2H, CH-2, CH-4), 1.80 (d, *J =* 12.8 Hz, 1H, CH_a_-3), 1.07 (t, *J =* 7.1 Hz, 3H, CH_3_), 0.87 (d, *J =* 6.6 Hz, 6H, CH_3_-6, CH_3_-7), 0.67 (q, *J =* 12.4 Hz, 1H, CH_b_-3). ^13^C-NMR (DMSO-*d*_6_): δ 169.60, 158.68, 156.31, 152.59, 141.88, 136.13, 134.17, 129.20, 122.57, 120.34 (2C), 120.33, 119.25, 115.38 (2C), 60.47, 59.71 (2C), 45.02, 41.61, 33.79, 30.90 (2C), 19.32 (2C), 13.92. HRMS (ESI) *m*/*z*: [M + H]^+^ calculated for C_26_H_32_O_3_N_3_F_4_, 510.23743; found, 510.23688, Δ −1.08 ppm.

#### 3.1.43. Preparation of Racemic 3-{4-(3,5-dimethylpiperidin-1-yl)-3-[3-(4-fluorophenyl)ureido]-phenyl}-4,4,4-trifluorobutyric Acid (**i18**)

Hydrolysis of compound **g18** (456 mg, 0.89 mmol) following the general procedure B afforded compound **i18** (white solid, 371 mg, 86.3% yield). mp: 125.1–126.4 °C. ^1^H-NMR (DMSO-*d*_6_): δ 12.50 (s, 1H, COOH), 9.76 (s, 1H, NH), 8.14 (d, *J =* 1.8 Hz, 1H, H-phenyl), 8.11 (s, 1H, NH), 7.55–7.48 (m, 2H, H-phenyl), 7.16–7.09 (m, 3H, H-phenyl), 6.98 (dd, *J =* 8.2, 1.9 Hz, 1H, H-phenyl), 3.96–3.79 (m, 1H, CH), 2.99–2.74 (m, 4H, CH_2_, CH_a_-1 and CH_a_-5), 2.14 (td, *J =* 11.0, 2.8 Hz, 2H, CH_b_-1 and CH_b_-5), 2.08–1.93 (m, 2H, CH-2, CH-4), 1.79 (d, *J =* 12.8 Hz, 1H, CH_a_-3), 0.86 (d, *J =* 6.5 Hz, 6H, CH_3_-6, CH_3_-7), 0.66 (q, *J =* 12.4 Hz, 1H, CH_b_-3). HRMS (ESI) *m*/*z*: [M + H]^+^ calculated for C_24_H_28_O_3_N_3_F_4_, 482.20613; found, 482.20630, Δ 0.35 ppm.

#### 3.1.44. Preparation of Racemic 3-[3-[3-(4-bromophenyl)ureido]-4-(3,5-dimethylpiperidin-1-yl)-phenyl]-4,4,4-trifluorobutyric Acid Ethyl Ester (**g19**)

Reaction of compound **f** and 4-bromophenyl isocyanate following the general procedure A afforded compound **g19** (white solid, 100 mg, 81.5% yield). ^1^H-NMR (DMSO-*d*_6_): δ 9.69 (s, 1H, NH), 8.13 (d, *J =* 1.7 Hz, 1H, H-phenyl), 8.09 (s, 1H, NH), 7.50–7.44 (m, 4H, H-phenyl), 7.14 (d, *J =* 8.2 Hz, 1H, H-phenyl), 7.00 (dd, *J =* 8.2, 1.8 Hz, 1H, H-phenyl), 4.07–3.87 (m, 3H, CH_2_, CH), 3.07–2.80 (m, 4H, CH_2_, CH_a_-1 and CH_a_-5), 2.14 (t, *J =* 10.7 Hz, 2H, CH_b_-1 and CH_b_-5), 2.05–1.91 (m, 2H, CH-2, CH-4), 1.80 (d, *J =* 12.6 Hz, 1H, CH_a_-3), 1.07 (t, *J =* 7.1 Hz, 3H, CH_3_), 0.87 (d, *J =* 6.6 Hz, 6H, CH_3_-6, CH_3_-7), 0.67 (q, *J =* 12.0 Hz, 1H, CH_b_-3). HRMS (ESI) *m*/*z*: [M + H]^+^ calculated for C_26_H_33_O_3_N_3_BrF_3_, 570.15737; found, 570.15735, Δ −0.03 ppm.

#### 3.1.45. Preparation of Racemic 3-[3-[3-(4-bromophenyl)ureido]-4-(3,5-dimethylpiperidin-1-yl)-phenyl]-4,4,4-trifluorobutyric Acid (**i19**)

Hydrolysis of compound **g19** (100 mg, 0.18 mmol) following the general procedure B afforded compound **i19** (white solid, 71 mg, 74.8% yield). mp: 136.9–137.7 °C. ^1^H-NMR (DMSO-*d*_6_): δ 12.58 (s, 1H, COOH), 9.75 (s, 1H, -NH-), 8.15–8.09 (m, 2H, H-phenyl, NH), 7.51–7.42 (m, 4H, H-phenyl), 7.14 (d, *J =* 8.2 Hz, 1H, H-phenyl), 6.98 (dd, *J =* 8.1, 1.6 Hz, 1H, H-phenyl), 3.96–3.82 (m, 1H, CH), 2.98–2.71 (m, 4H, CH_2_, CH_a_-1 and CH_a_-5), 2.14 (td, *J =* 11.0, 3.3 Hz, 2H, CH_b_-1 and CH_b_-5), 2.06–1.91 (m, 2H, CH-2, CH-4), 1.80 (d, *J =* 13.7 Hz, 1H, CH_a_-3), 0.86 (d, *J =* 6.5 Hz, 6H, CH_3_-6, CH_3_-7), 0.66 (q, *J =* 12.4 Hz, 1H, CH_b_-3). ^13^C-NMR (DMSO-*d*_6_): δ 171.14, 152.47, 141.95, 139.41, 134.10, 131.57 (2C), 129.57, 126.92, 122.81, 120.42, 120.34 (2C), 119.36, 113.30, 59.79 (2C), 45.20, 41.68, 33.98, 30.79 (2C), 19.34 (2C). HRMS (ESI) *m*/*z*: [M + H]^+^ calculated for C_24_H_28_O_3_N_3_BrF_3_, 542.12607; found, 542.12616, Δ 0.17 ppm.

#### 3.1.46. Preparation of Racemic 3-{4-(3,5-dimethylpiperidin-1-yl)-3-[3-(4-ethylphenyl)ureido]phenyl}-4,4,4-trifluorobutyric Acid Ethyl Ester (**g20**)

Reaction of compound **f** and 4-ethylphenethyl isocyanate following the general procedure A afforded compound **g20** (white solid, 83.7% yield). ^1^H-NMR (DMSO-*d*_6_): δ 9.42 (s, 1H, NH), 8.15 (d, *J =* 1.9 Hz, 1H, H-phenyl), 8.03 (s, 1H, NH), 7.41–7.37 (m, 2H, H-phenyl), 7.16–7.10 (m, 3H, H-phenyl), 6.97 (dd, *J =* 8.2, 2.0 Hz, 1H, H-phenyl), 4.08–3.86 (m, 3H, CH_2_, CH), 3.06–2.78 (m, 4H, CH_2_, CH_a_-1 and CH_a_-5), 2.55 (q, *J =* 7.6 Hz, 2H, CH_3_CH_2_-phenyl), 2.13 (td, *J =* 11.2, 2.3 Hz, 2H, CH_b_-1 and CH_b_-5), 2.01–1.87 (m, 2H, CH-2, CH-4), 1.78 (d, *J =* 12.8 Hz, 1H, CH_a_-3), 1.16 (t, *J =* 7.6 Hz, 3H, CH_3_CH2-phenyl), 1.07 (t, *J =* 7.6 Hz, 3H, CH_3_), 0.86 (d, *J =* 6.6 Hz, 6H, CH_3_-6, CH_3_-7), 0.66 (q, *J =* 12.4 Hz, 1H, CH_b_-3). HRMS (ESI) *m*/*z*: [M + H]^+^ calculated for C_28_H_37_O_3_N_3_F_3_, 520.27815; found, 520.27795, Δ −0.39 ppm.

#### 3.1.47. Preparation of Racemic 3-{4-(3,5-dimethylpiperidin-1-yl)-3-[3-(4-ethylphenyl)ureido]phenyl}-4,4,4-trifluorobutyric Acid (**i20**)

Hydrolysis of compound **g20** (134 mg, 0.26 mmol) following the general procedure B afforded compound **i20** (white solid, 103 mg, 81.2% yield). mp: 122.1–122.9 °C. ^1^H-NMR (DMSO-*d*_6_): δ 9.44 (s, 1H, NH), 8.14 (d, *J =* 1.6 Hz, 1H, H-phenyl), 8.04 (s, 1H, NH), 7.39 (d, *J =* 8.4 Hz, 2H, H-phenyl), 7.16–7.10 (m, 3H, H-phenyl), 6.96 (dd, *J =* 8.2, 1.5 Hz, 1H, H-phenyl), 3.95–3.81 (m, 1H, CH), 2.96–2.73 (m, 4H, CH_2_, CH_a_-1 and CH_a_-5), 2.55 (q, *J =* 7.6 Hz, 2H, CH_3_CH_2_-phenyl), 2.13 (td, *J =* 11.0, 3.8 Hz, 2H, CH_b_-1 and CH_b_-5), 2.00–1.88 (m, 2H, CH-2, CH-4), 1.78 (d, *J =* 12.7 Hz, 1H, CH_a_-3), 1.16 (t, *J =* 7.6 Hz, 3H, CH_3_CH2-phenyl), 0.86 (d, *J =* 6.6 Hz, 6H, CH_3_-6, CH_3_-7), 0.65 (q, *J =* 12.4 Hz, 1H, CH_b_-3). ^13^C-NMR (DMSO-*d*_6_): δ 171.21, 152.63, 141.72, 137.54, 137.40, 134.33, 129.60, 128.10 (2C), 126.93, 122.49, 120.36, 119.19, 118.97 (2C), 59.74 (2C), 45.18, 41.64, 34.00, 30.86 (2C), 27.60, 19.32 (2C), 15.85. HRMS (ESI) *m*/*z*: [M + H]^+^ calculated for C_26_H_33_O_3_N_3_F_3_, 492.24685; found, 492.24774, Δ 1.80 ppm.

#### 3.1.48. Preparation of Racemic 3-{4-(3,5-dimethylpiperidin-1-yl)-3-[3-(4-isopropylphenyl)ureido]-phenyl}-4,4,4-trifluorobutyric Acid Ethyl Ester (**g21**)

Reaction of compound **f** and 4-isopropylphenyl isocyanate following the general procedure A afforded compound **g21** (white solid, 80.7% yield). ^1^H-NMR (DMSO-*d*_6_): δ 9.42 (s, 1H, NH), 8.15 (d, *J =* 1.9 Hz, 1H, H-phenyl), 8.03 (s, 1H, NH), 7.42–7.37 (m, 2H, H-phenyl), 7.14–7.20 (m, 2H, H-phenyl), 7.12 (d, *J =* 8.4 Hz, 1H, H-phenyl), 6.97 (dd, *J =* 8.2, 1.9 Hz, 1H, H-phenyl), 4.08–3.86 (m, 3H, CH_2_, CH), 3.07–2.79 (m, 5H, CH_2_, CH_a_-1 and CH_a_-5, (CH_3_)_2_CH-phenyl), 2.13 (td, *J =* 11.1, 2.5 Hz, 2H, CH_b_-1 and CH_b_-5), 2.00–1.85 (m, 2H, CH-2, CH-4), 1.78 (d, *J =* 12.9 Hz, 1H, CH_a_-3), 1.18 (d, *J =* 6.8 Hz, 6H, (CH_3_)_2_CH-phenyl), 1.08 (t, *J =* 7.2 Hz, 3H, CH_3_), 0.86 (d, *J =* 6.6 Hz, 6H, CH_3_-6, CH_3_-7), 0.65 (q, *J =* 12.4 Hz, 1H, CH_b_-3). HRMS (ESI) *m*/*z*: [M + H]^+^ calculated for C_29_H_39_O_3_N_3_F_3_, 534.29380; found, 534.29602, Δ 4.15 ppm.

#### 3.1.49. Preparation of Racemic 3-{4-(3,5-dimethylpiperidin-1-yl)-3-[3-(4-isopropylphenyl)ureido]-phenyl}-4,4,4-trifluorobutyric Acid (**i21**)

Hydrolysis of compound **g21** (200 mg, 0.35 mmol) following the general procedure B afforded compound **i21** (white solid, 142 mg, 81.4% yield). mp: 121.7–122.7 °C. ^1^H-NMR (DMSO-*d*_6_): δ 9.43 (s, 1H, NH), 8.15 (d, *J =* 1.6 Hz, 1H, H-phenyl), 8.04 (s, 1H, NH), 7.39 (d, *J =* 8.4 Hz, 2H, H-phenyl), 7.16 (d, *J =* 8.5 Hz, 2H, H-phenyl), 7.12 (d, *J =* 8 Hz, 1H, H-phenyl), 6.96 (dd, *J =* 8.3, 1.7 Hz, 1H, H-phenyl), 3.95–3.82 (m, 1H, CH), 2.98–2.73 (m, 5H, CH_2_, CH_a_-1 and CH_a_-5, (CH_3_)_2_CH-phenyl), 2.13 (td, *J =* 11.0, 3.9 Hz, 2H, CH_b_-1 and CH_b_-5), 2.00–1.85 (m, 2H, CH-2, CH-4), 1.77 (d, *J =* 12.7 Hz, 1H, CH_a_-3), 1.18 (t, *J =* 7.1 Hz, 6H, (CH_3_)_2_CH-phenyl), 0.85 (d, *J =* 6.6 Hz, 6H, CH_3_-6, CH_3_-7), 0.65 (q, *J =* 12.4 Hz, 1H, CH_b_-3). ^13^C-NMR (DMSO-*d*_6_): δ 171.13, 152.64, 142.30, 141.72, 137.44, 134.34, 129.56, 126.92, 126.62 (2C), 122.47, 120.37, 119.18, 119.09 (2C), 59.73 (2C), 45.19, 41.63, 33.92, 32.87, 30.87 (2C), 24.09 (3C), 19.32 (2C). HRMS (ESI) *m*/*z*: [M + H]^+^ calculated for C_27_H_35_O_3_N_3_F_3_, 506.26250; found, 506.26373, Δ 2.42 ppm.

#### 3.1.50. Preparation of Racemic 3-{4-(3,5-dimethylpiperidin-1-yl)-3-[3-(4-trifluoromethoxyphenyl)-ureido]phenyl}-4,4,4-trifluorobutyric Acid Ethyl Ester (**g22**)

Reaction of compound **f** and 4-fluorophenyl isocyanate following the general procedure A afforded compound **g22** (white solid, 81.5% yield). ^1^H-NMR (DMSO-*d*_6_): δ 9.75 (s, 1H, NH), 8.14 (d, *J =* 1.9 Hz, 1H, H-phenyl), 8.10 (s, 1H, NH), 7.62–7.58 (m, 2H, H-phenyl), 7.30 (d, *J =* 8.5 Hz, 2H, H-phenyl), 7.14 (d, *J =* 8.2 Hz, 1H, H-phenyl), 7.00 (dd, *J =* 8.2, 1.9 Hz, 1H, H-phenyl), 4.07–3.88 (m, 3H, CH_2_, CH), 3.07–2.81 (m, 4H, CH_2_, CH_a_-1 and CH_a_-5), 2.15 (t, *J =* 10.4 Hz, 2H, CH_b_-1 and CH_b_-5), 2.05–1.90 (m, 2H, CH-2, CH-4), 1.80 (d, *J =* 12.8 Hz, 1H, CH_a_-3), 1.07 (t, *J =* 7.1 Hz, 3H, CH_3_), 0.87 (d, *J =* 6.6 Hz, 6H, CH_3_-6, CH_3_-7), 0.67 (q, *J =* 12.8 Hz, 1H, CH_b_-3). HRMS (ESI) *m*/*z*: [M + H]^+^ calculated for C_27_H_32_O_4_N_3_F_6_, 576.22915; found, 576.22772, Δ −2.49 ppm.

#### 3.1.51. Preparation of Racemic 3-{4-(3,5-dimethylpiperidin-1-yl)-3-[3-(4-trifluoromethoxyphenyl)-ureido]phenyl}-4,4,4-trifluorobutyric Acid (**i22**)

Hydrolysis of compound **g22** (586 mg, 1.02 mmol) following the general procedure B afforded compound **i22** (white solid, 455 mg, 81.5% yield). mp: 196.2–197.8 °C. ^1^H-NMR (DMSO-*d*_6_): δ 12.37 (s, 1H, COOH), 9.78 (s, 1H, NH), 8.17–8.08 (m, 2H, H-phenyl, NH), 7.64–7.56 (m, 2H, H-phenyl), 7.30 (d, *J =* 8.7 Hz, 2H, H-phenyl), 7.15 (d, *J =* 8.2 Hz, 1H, H-phenyl), 7.00 (dd, *J =* 8.2, 1.5 Hz, 1H, H-phenyl), 3.97–3.82 (m, 1H, -CH-), 3.01–2.74 (m, 4H, CH_2_, CH_a_-1 and CH_a_-5), 2.15 (td, *J =* 11.0, 2.8 Hz, 2H, CH_b_-1 and CH_b_-5), 2.07–1.91 (m, 2H, CH-2, CH-4), 1.80 (d, *J =* 12.6 Hz, 1H, CH_a_-3), 0.87 (d, *J =* 6.5 Hz, 6H, CH_3_-6, CH_3_-7), 0.67 (q, *J =* 12.4 Hz, 1H, CH_b_-3). ^13^C-NMR (DMSO-*d*_6_): δ 171.02, 152.47, 142.73, 141.93, 139.14, 134.00, 129.51, 126.86, 122.87, 121.79 (2C), 120.44, 120.28, 119.65 (2C), 119.32, 59.74 (2C), 45.09, 41.62, 33.77, 30.89 (2C), 19.32 (2C). HRMS (ESI) *m*/*z*: [M + H]^+^ calculated for C_25_H_28_O_4_N_3_F_6_, 548.19785; found, 548.19781, Δ −0.08 ppm.

#### 3.1.52. Preparation of Racemic 3-{4-(3,5-dimethylpiperidin-1-yl)-3-[3-(4-trifluoromethylphenyl)-ureido]phenyl}-4,4,4-trifluorobutyric Acid Ethyl Ester (**g23**)

Reaction of compound **f** and 4-(trifluoromethyl)phenyl isocyanate following the general procedure A afforded compound **g23** (white solid, 80.6% yield). ^1^H-NMR (DMSO-*d*_6_): δ 9.96 (s, 1H, NH), 8.26–8.12 (m, 2H, NH, H-phenyl), 7.81–7.61 (m, 4H, H-phenyl), 7.16 (d, *J =* 8.2 Hz, 1H, H-phenyl), 7.02 (d, *J =* 8.2 Hz, 1H, H-phenyl), 4.12–3.89 (m, 3H, CH_2_, CH), 3.13–2.79 (m, 4H, CH_2_, CH_a_-1 and CH_a_-5), 2.15 (t, *J =* 10.6 Hz, 2H, CH_b_-1 and CH_b_-5), 2.08–1.92 (m, 2H, CH-2, CH-4), 1.81 (d, *J =* 12.1 Hz, 1H, CH_a_-3), 1.14–1.03 (m, 3H, CH_3_), 0.87 (d, *J =* 6.4 Hz, 6H, CH_3_-6, CH_3_-7), 0.67 (q, *J =* 12.0 Hz, 1H, CH_b_-3). HRMS (ESI) *m*/*z*: [M + H]^+^ calculated for C_27_H_32_O_3_N_3_F_6_, 560.23424; found, 560.23425, Δ 0.02 ppm.

#### 3.1.53. Preparation of Racemic 3-{4-(3,5-dimethylpiperidin-1-yl)-3-[3-(4-trifluoromethylphenyl)-ureido]phenyl}-4,4,4-trifluorobutyric Acid (**i23**)

Hydrolysis of compound **g23** (350 mg, 0.63 mmol) following the general procedure B afforded compound **i23** (white solid, 270 mg, 80.6% yield). mp: 179.7–180.5 °C. ^1^H-NMR (DMSO-*d*_6_): δ 12.50 (s, 1H, COOH), 9.97 (s, 1H, NH), 8.19 (s, 1H, NH), 8.14 (d, *J =* 1.9 Hz, 1H, H-phenyl), 7.74–7.62 (m, 4H, H-phenyl), 7.16 (d, *J =* 8.2 Hz, 1H, H-phenyl), 7.01 (dd, *J =* 8.3, 2.0 Hz, 1H, H-phenyl), 3.98–3.82 (m, 1H, CH), 3.00–2.76 (m, 4H, CH_2_, CH_a_-1 and CH_a_-5), 2.16 (td, *J =* 11.1, 3.1 Hz, 2H, CH_b_-1 and CH_b_-5), 2.07–1.92 (m, 2H, CH-2, CH-4), 1.81 (d, *J =* 12.8 Hz, 1H, CH_a_-3), 0.87 (d, *J =* 6.6 Hz, 6H, CH_3_-6, CH_3_-7), 0.67 (q, *J =* 12.4 Hz, 1H, CH_b_-3). HRMS (ESI) *m*/*z*: [M + H]^+^ calculated for C_25_H_28_O_3_N_3_F_6_, 532.20294; found, 532.20508, Δ 4.03 ppm.

#### 3.1.54. Preparation of Racemic 3-{4-(3,5-dimethylpiperidin-1-yl)-3-[3-(4-nitrophenyl)ureido]phenyl}-4,4,4-trifluorobutyric Acid Ethyl Ester (**g24**)

Reaction of compound **f** and 4-nitrophenyl isocyanate following the general procedure A afforded compound **g24** (yellow solid, 86.5% yield). ^1^H-NMR (DMSO-*d*_6_): δ 10.28 (s, 1H, NH), 8.29 (s, 1H, NH), 8.24–8.19 (m, 2H, H-phenyl), 8.15 (d, *J =* 1.8 Hz, 1H, H-phenyl), 7.78–7.73 (m, 2H, H-phenyl), 7.17 (d, *J =* 8.2 Hz, 1H, H-phenyl), 7.04 (dd, *J =* 8.2, 1.9 Hz, 1H, H-phenyl), 4.07–3.91 (m, 3H, CH_2_, CH), 3.08–2.82 (m, 4H, CH_2_, CH_a_-1 and CH_a_-5), 2.15 (dd, *J =* 16.1, 6.1 Hz, 2H, CH_b_-1 and CH_b_-5), 2.07–1.92 (m, 2H, CH-2, CH-4), 1.81 (d, *J =* 12.8 Hz, 1H, CH_a_-3), 1.08 (t, *J =* 7.1 Hz, 3H, CH_3_), 0.87 (d, *J =* 6.6 Hz, 6H, CH_3_-6, CH_3_-7), 0.67 (q, *J =* 12.4 Hz, 1H, CH_b_-3). HRMS (ESI) *m*/*z*: [M + H]^+^ calculated for C_26_H_32_O_5_N_4_F_3_, 537.23193; found, 537.23126, Δ −1.25 ppm.

#### 3.1.55. Preparation of Racemic 3-{4-(3,5-dimethylpiperidin-1-yl)-3-[3-(4-nitrophenyl)ureido]phenyl}-4,4,4-trifluorobutyric Acid (**i24**)

Hydrolysis of compound **g24** (160 mg, 0.30 mmol) following the general procedure B afforded compound **i24** (yellow solid, 121 mg, 79.3% yield). mp: 151.3–153.0 °C. ^1^H-NMR (DMSO-*d*_6_): δ 12.51 (s, 1H, COOH), 10.29 (s, 1H, NH), 8.29 (s, 1H, NH), 8.23–8.19 (m, 2H, H-phenyl), 8.14 (d, *J =* 1.6 Hz, 1H, H-phenyl), 7.77–7.72 (m, 2H, H-phenyl), 7.17 (d, *J =* 8.3 Hz, 1H, H-phenyl), 7.04 (dd, *J =* 8.2, 1.7 Hz, 1H, H-phenyl), 3.98–3.85 (m, 1H, CH), 3.01–2.77 (m, 4H, CH_2_, CH_a_-1 and CH_a_-5), 2.16 (td, *J =* 11.0, 2.8 Hz, 2H, CH_b_-1 and CH_b_-5), 2.08-1.93 (m, 2H, CH–2, CH-4), 1.81 (d, *J =* 12.8 Hz, 1H, CH_a_-3), 0.86 (t, *J =* 8.4 Hz, 6H, CH_3_-6, CH_3_-7), 0.68 (q, *J =* 12.4 Hz, 1H, CH_b_-3). ^13^C-NMR (DMSO-*d*_6_): δ 171.00, 152.05, 146.58, 142.19, 141.09, 133.52, 129.62, 126.84, 125.24 (2C), 123.49, 120.59, 119.65, 117.62 (2C), 59.79 (2C), 45.04, 41.59, 33.74, 30.80 (2C), 19.32 (2C). HRMS (ESI) *m*/*z*: [M + H]^+^ calculated for C_24_H_28_O_5_N_4_F_3_, 509.20063; found, 509.20267, Δ 4.00 ppm.

#### 3.1.56. Preparation of Racemic 3-[3-[2-(4-cyanophenyl)acetylamino]-4-(3,5-dimethylpiperidin-1-yl)-phenyl]-4,4,4-trifluorobutyric Acid Ethyl Ester (**h1**)

To a solution of racemic 3-[3-amino-4-(3,5-dimethylpiperidin-1-yl)-phenyl]-4,4,4-trifluoro-butyric acid ethyl ester (**f**, 120 mg, 0.32 mmol) in *N*,*N*-dimethylformamide (4 mL) was added 4-cyanophenylacetic acid (52 mg, 0.32 mmol), 1-hydroxybenzotriazole (43 mg, 0.32 mmol), *N*-(3-dimethylaminopropyl)-*N*’-ethylcarbodiimide hydrochloride (61 mg, 0.32 mmol) and *N*,*N*-diisopropylethylamine (106 μL, 0.64 mmol). The reaction mixture was stirred at room temperature until the starting material disappeared in TLC. The reaction mixture was then diluted with ethyl acetate (100 mL), washed with 1 N HCl aqueous solution (40 mL × 2), saturated NaCl aqueous solution (40 mL × 2), dried over anhydrous Na_2_SO_4_ and concentrated. The residue was purified by column chromatography (silica gel, PE/EA = 10:1, *v*/*v*) to afford the product as a colorless oil (117 mg, 70.5% yield). ^1^H-NMR (DMSO-*d*_6_): δ 8.85 (s, 1H, NH), 8.04 (s, 1H, H-phenyl), 7.84 (d, *J =* 8.1 Hz, 2H, H-phenyl), 7.59 (d, *J =* 8.1 Hz, 2H, H-phenyl), 7.11 (d, *J =* 8.0 Hz, 2H, H-phenyl), 4.05–3.86 (m, 5H, CH_2_, CH, CH_2_-phenyl), 3.06–2.82 (m, 2H, CH_2_), 2.73–2.63 (m, 2H, CH_a_-1 and CH_a_-5), 2.04 (t, *J =* 11.1 Hz, 2H, CH_b_-1 and CH_b_-5), 1.67 (d, *J =* 12.0 Hz, 1H, CH_a_-3), 1.61–1.45 (m, 2H, CH-2, CH-4), 1.05 (t, *J =* 7.1 Hz, 3H, CH_3_), 0.77 (d, *J =* 6.6 Hz, 6H, CH_3_-6, CH_3_-7), 0.57 (q, *J =* 12.0 Hz, 1H, CH_b_-3). HRMS (ESI) *m*/*z*: [M + H]^+^ calculated for C_28_H_33_O_3_N_3_F_3_, 516.24685; found, 516.24878, Δ 3.73 ppm.

#### 3.1.57. Preparation of Racemic 3-[3-[2-(4-cyanophenyl)acetylamino]-4-(3,5-dimethylpiperidin-1-yl)-phenyl]-4,4,4-trifluorobutyric Acid (**j1**)

Hydrolysis of compound **h1** (70 mg, 0.14 mmol) following the general procedure B afforded compound **j1** (white solid, 49 mg, 73.9% yield). mp: 108.2–108.9 °C. ^1^H-NMR (DMSO-*d*_6_): δ 8.83 (s, 1H), 8.07 (s, 1H, H-phenyl), 7.84 (d, *J =* 6.1 Hz, 2H, H-phenyl), 7.59 (d, *J =* 6.6 Hz, 2H, H-phenyl), 7.13–7.03 (m, 2H, H-phenyl), 3.98–3.82 (m, 3H, CH, CH_2_-phenyl), 2.91–2.61 (m, 4H, CH_2_, CH_a_-1 and CH_a_-5), 2.03 (t, *J =* 11.0 Hz, 2H, CH_b_-1 and CH_b_-5), 1.68 (d, *J =* 12.4 Hz, 1H, CH_a_-3), 1.59–1.43 (m, 2H, CH-2, CH-4), 0.79–0.68 (m, 6H, CH_3_-6, CH_3_-7), 0.57 (q, *J =* 12.0 Hz, 1H, CH_b_-3). ^13^C-NMR (DMSO-*d*_6_): δ 168.08, 143.22, 141.42, 132.58 (2C), 132.40, 130.58 (2C), 129.58, 127.02, 125.04, 121.43, 120.35, 118.84, 109.88, 59.35 (2C), 45.16, 43.55, 41.42, 34.36, 31.00 (2C), 19.16 (2C). HRMS (ESI) *m*/*z*: [M + H]^+^ calculated for C_26_H_29_O_3_N_3_F_3_, 488.21555; found, 488.21490, Δ −1.34 ppm.

#### 3.1.58. Preparation of Racemic 3-{4-(3,5-dimethylpiperidin-1-yl)-3-[2-(4-trifluoromethylphenyl)-acetylamino]phenyl}-4,4,4-trifluorobutyric Acid Ethyl Ester (**h2**)

Reaction of compound **f** and 4-(trifluoromethyl)phenylacetic acid following the similar procedure described for the preparation of **h1** afforded **h2** as a colorless oil (128 mg, 71.2%). ^1^H-NMR (DMSO-*d*_6_): δ 8.78 (s, 1H, NH), 8.12 (s, 1H, H-phenyl), 7.74 (d, *J =* 8.2 Hz, 2H, H-phenyl), 7.63 (d, *J =* 8.2 Hz, 2H, H-phenyl), 7.10 (s, 2H, H-phenyl), 4.05–3.92 (m, 3H, CH, CH_2_), 3.90 (s, 2H, CH_2_-phenyl), 3.06–2.82 (m, 2H, CH_2_), 2.59–2.69 (m, 2H, CH_a_-1 and CH_a_-5), 2.01 (t, *J =* 11.1 Hz, 2H, CH_b_-1 and CH_b_-5), 1.62 (d, *J =* 13.0 Hz, 1H, CH_a_-3), 1.47-1.34 (dt, *J =* 22.8, 11.6 Hz, 2H, CH-2, CH-4), 1.06 (t, *J =* 7.1 Hz, 3H, CH_3_), 0.73 (d, *J =* 6.6 Hz, 6H, CH_3_-6, CH_3_-7), 0.53 (q, *J =* 12.4 Hz, 1H, CH_b_-3). HRMS (ESI) *m*/*z*: [M + H]^+^ calculated for C_28_H_33_O_3_N_2_F_6_, 559.23899; found, 559.24072, Δ 3.10 ppm.

#### 3.1.59. Preparation of Racemic 3-{4-(3,5-dimethylpiperidin-1-yl)-3-[2-(4-trifluoromethylphenyl)-acetylamino]phenyl}-4,4,4-trifluorobutyric Acid (**j2**)

Hydrolysis of compound **h2** (80 mg, 0.14 mmol) following the general procedure B afforded compound **j2** (white solid, 49 mg, 77.8% yield). mp: 93.2–93.6 °C. ^1^H-NMR (DMSO-*d*_6_): δ 8.77 (s, 1H, NH), 8.13 (s, 1H, H-phenyl), 7.74 (d, *J* = 8.1 Hz, 2H, H-phenyl), 7.63 (d, *J* = 8.0 Hz, 2H, H-phenyl), 7.13–7.02 (m, 2H, H-phenyl), 3.98–3.82 (m, 3H, CH, CH_2_), 2.90–2.58 (m, 4H, CH_2_, CH_a_-1 and CH_a_-5), 2.01 (td, *J* = 11.1, 4.2 Hz, 2H, CH_b_-1 and CH_b_-5), 1.61 (d, *J* = 12.6 Hz, 1H, CH_a_-3), 1.46–1.30 (m, 2H, CH-2, CH-4), 0.73 (d, *J* = 6.6 Hz, 6H, CH_3_-6, CH_3_-7), 0.53 (q, *J =* 12.0 Hz, 1H, CH_b_-3). ^13^C-NMR (DMSO-*d*_6_): δ 171.22, 168.28, 143.04, 140.23, 132.59, 130.44 (2C), 129.50, 127.98-125.48 (4C), 124.89, 123.05, 120.92, 120.49, 59.36 (2C), 45.02, 43.47, 41.29, 33.96, 30.91 (2C), 19.07 (2C). HRMS (ESI) *m*/*z*: [M + H]^+^ calculated for C_26_H_29_O_3_N_2_F_6_, 531.20769; found, 531.20703, Δ −1.24 ppm.

#### 3.1.60. Preparation of Racemic 3-{4-(3,5-dimethylpiperidin-1-yl)-3-[2-(4-nitrophenyl)acetylamino]-phenyl}-4,4,4-trifluorobutyric Acid Ethyl Ester (**h3**)

Reaction of compound **f** and 4-nitrophenylacetic acid following the similar procedure described for the preparation of **h1** afforded **h3** as a light yellow oil (99 mg, 68.7%). ^1^H-NMR (DMSO-*d*_6_): δ 8.91 (s, 1H, NH), 8.24 (d, *J =* 8.6 Hz, 2H, H-phenyl), 8.03 (s, 1H, H-phenyl), 7.67 (d, *J =* 8.5 Hz, 2H, H-phenyl), 7.11 (s, 2H, H-phenyl), 4.05–3.88 (m, 5H, CH_2_, CH, CH_2_-phenyl), 3.06–1.99 (m, 2H, CH_2_), 2.75–2.65 (m, 2H, CH_a_-1 and CH_a_-5), 2.04 (t, *J =* 11.1 Hz, 2H, CH_b_-1 and CH_b_-5), 1.70–1.50 (m, 3H, CH_a_-3, CH-2, CH-4), 1.05 (t, *J =* 7.1 Hz, 3H, CH_3_), 0.75 (d, *J =* 6.6 Hz, 6H, CH_3_-6, CH_3_-7), 0.56 (q, *J =* 12.0 Hz, 1H, CH_b_-3). HRMS (ESI) *m*/*z*: [M + H]^+^ calculated for C_27_H_33_O_5_N_3_F_3_, 536.23668; found, 536.23749, Δ 1.51 ppm.

#### 3.1.61. Preparation of Racemic 3-{4-(3,5-dimethylpiperidin-1-yl)-3-[2-(4-nitrophenyl)acetylamino]-phenyl}-4,4,4-trifluorobutyric Acid (**j3**)

Hydrolysis of compound **h3** (70 mg, 0.13 mmol) following the general procedure B afforded compound **j3** (yellow solid, 52 mg, 78.2% yield). mp: 92.1–92.7 °C. ^1^H-NMR (DMSO-*d*_6_): δ 12.51 (s, 1H, COOH), 8.90 (s, 1H, NH), 8.24 (d, *J =* 8.7 Hz, 2H, H-phenyl), 8.04 (s, 1H, H-phenyl), 7.67 (d, *J =* 8.6 Hz, 2H, H-phenyl), 7.11 (s, 2H, H-phenyl), 3.99–3.84 (m, 3H, CH_2_-phenyl, CH), 2.94–2.64 (m, 4H, CH_2_, CH_a_-1 and CH_a_-5), 2.04 (t, *J =* 9.9 Hz, 2H, CH_b_-1 and CH_b_-5), 1.69–1.48 (m, 3H, CH_a_-3, CH-2, CH-4), 0.75 (d, *J =* 6.6 Hz, 6H, CH_3_-6, CH_3_-7), 0.56 (q, *J =* 12.4 Hz, 1H, CH_b_-3). ^13^C-NMR (DMSO-*d*_6_): δ 170.95, 168.00, 146.64, 143.69, 132.37, 130.95, 130.83 (2C), 129.17, 126.78, 125.12, 123.73 (2C), 121.62, 120.45, 59.34 (2C), 45.85, 43.20, 41.37, 33.66, 30.96 (2C), 19.09 (2C). HRMS (ESI) *m*/*z*: [M + H]^+^ calculated for C_25_H_29_O_5_N_3_F_3_, 508.20538; found, 508.20618, Δ 1.57 ppm.

### 3.2. Pharmacological Methods

#### 3.2.1. The Enzyme Assay for IDO1 and TDO Inhibition

Recombinant human IDO1 and TDO were expressed and purified according to the reported procedures [20]. The assay was performed according to the literature: A standard reaction mixture (100 µL) containing 100 mM potassium phosphate buffer (pH 6.5), 40 mM ascorbic acid (neutralized with NaOH), 200 µg/mL catalase, 20 µM methylene blue and 0.05 µM rhIDO1 or rhTDO was added to the solution containing the substrate L-tryptophan and the test sample at a determined concentration. The reaction was carried out at 37 °C for 45 min and stopped by adding 20 µL of 30% (*w*/*v*) trichloroacetic acid. After heating at 65 °C for 15 min, 100 µL of 2% (*w*/*v*) p-dimethylaminobenzaldehyde in acetic acid was added to each well. The yellow pigment derived from kynurenine was measured at 492 nm using a SYNERGY-H1 microplate reader (Biotek Instruments, Inc., Winooski, VT, USA). IC_50_ was analyzed using the GraphPad Prism 8.0 software (GraphPad Software, San Diego, CA, USA) [21].

#### 3.2.2. Mice

C57BL/6 mice were obtained from Beijing Vital River Laboratory Animal Technology Co., Ltd. (Beijing, China). Studies involving mice were approved by the Experimental Animal Management and Welfare Committee at the Institute of Materia Medica, Peking Union Medical College (Approval No. 00000522).

#### 3.2.3. Pharmacokinetic Studies

The animal Care and Welfare Committee of Institute of Materia Medica, Chinese Academy of Medical Sciences approved all animal care, housing, and laboratory procedures. Male C57BL/6 mice were used in the single dose pharmacokinetic studies. Compound **i12** was prepared as a 3 mg/mL suspension with 0.5% CMC for oral use and was formulated as a 3 mg/mL solution with 10% DMSO in 20% HP-β-CD for intravenous injection. Sixteen mice were divided into two groups, 10 in the oral group and six in the intravenous group. After fasting 12 h with free access to water, mice were treated with a 3 mg/kg i.v. or 30 mg/kg oral dose of compound i12. Blood samples (50 µL) were collected at 5, 15, 30 min, 1, 2, 4, 6, 8, 12 and 24 h after oral administration and 2, 5, 15, 30 min, 1, 2, 4, 6, 8, 12 and 24 h after intravenous injection. After centrifugation, the plasma samples (20 µL) were precipitated by four volumes of acetonitrile. The supernatant were analyzed by liquid chromatography/tandem mass spectrometry with a Zorbax C18 column (50 mm × 2.1 mm, 3.5 µm). Compound detection was performed with the mass spectrometer in positive ionization mode by t-SIM: *m*/*z* 489.209 for compound **i12**. The pharmacokinetic parameters were calculated with WinNonlin software V6.3 using non-compartmental analysis (Pharsight Corporation, Mountain View, CA, USA).

#### 3.2.4. In Vivo Studies

The mouse melanoma cells B16F10 were cultured and harvested in saline at 6 × 10^6^ cells/0.2 mL volume. Cells (0.2 mL) were injected subcutaneously into male C57BL/6 mice at day 0 of the experiment, and treatment was initiated at day 1 following the mice enrolled randomly in control and experimental groups. For control group, 0.5% CMCNa was orally administered every day. The CTX group were administered CTX intraperitoneally at the dose of 100 mg/kg. Compound **i12** was dissolved in 0.5% CMC-Na for oral treatment. After 17 days, the mice were sacrificed and the tumors were stripped and weighted. The tumor growth inhibition (TGI) was calculated as TGI = (1 − tumor weight _treatment_/tumor weight _vehicle_) × 100%. The statistical analysis was performed with GraphPad Prism 8.0 software and the significance level was evaluated with one-way ANOVA model [22].

The mouse pancreatic cancer cells PAN02 were cultured and harvested in saline at 6 × 10^6^ cells/0.2 mL volume. Cells (0.2 mL) were injected subcutaneously into male C57BL/6 mice at day 0 of the experiment, and treatment was initiated at day 1 following the mice enrolled randomly in control and experimental groups. For control group, 0.5% CMCNa was orally administered every day. The CTX group were administered CTX intraperitoneally at the dose of 60 mg/kg Compound **i12** was dissolved in 0.5% CMC-Na for oral treatment. After 15 days, the mice were sacrificed and the tumors were stripped and weighted. The tumor growth inhibition (TGI) was calculated as TGI = (1 − tumor weight _treatment_/tumor weight _vehicle_) × 100%. The statistical analysis was performed with GraphPad Prism 8.0 software and the significance level was evaluated with one-way ANOVA model [22].

## 4. Conclusions

In summary, we designed a new series of phenyl urea derivatives as IDO1 inhibitors through a ring formation strategy. Systematic SAR led to the discovery of the promising anticancer compound **i12** with favourable drug-like properties. Compound **i12** had potent IDO1 inhibitory activity with the IC_50_ of 0.331 μM and exhibited a satisfactory PK profile with moderate plasma clearance (22.45 mL/min/kg) and high oral bioavailability (87.4%). In addition, Compound **i12** orally administered at 15 mg/kg daily showed a TGI of 40.5% in a B16F10 subcutaneous xenograft model and at 30 mg/kg daily showed a TGI of 34.3% in a PAN02 subcutaneous xenograft model. Overall, compound **i12** is a promising anti-tumor agent with the potent in vitro enzymatic activity, good pharmacokinetic properties and satisfied in vivo anti-tumor efficacy.

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
