# Peer review of "Design, Synthesis and Biological Evaluation of Phenyl Urea Derivatives as IDO1 Inhibitors"

_molecules, 2020, doi:10.3390/molecules25061447_

Round 1

Reviewer 1 Report

In presented manuscript, the authors described design and synthesis a new series compounds as potential IDO1 inhibitors. In this study, they described synthesis of 59 new compounds. Then the inhibitory activities of new compounds were evaluated against IDO1. As a result of this systematic study, the authors were described compound i12 as promising anticancer compound with potential drug-like properties.

I recommend that presented manuscript can be accepted for publication providing that all issues raised are appropriately dealt with:

1) However, these new compounds were not well characterized. The authors did not provide 13C NMR spectra for new compounds. I recommend preparation of supporting materials with copies of 1H and 13C NMR spectra of synthesized compounds.

2) Line 63-67: Description on figure 2. is not correct. During this study the authors did not performed the ring synthesis. Design strategies could be described as introducing of rigid substituent.

3) Line 71: The synthesis of compound b is nucleophilic aromatic substitution!

4) Line 71: The synthesis of compound c is not well known procedure. Please provide reference for this transformation.

5) Line 84 and 85: Please provide yields for the step iv and v.

6) Line 241-249: The preparation and characterization of compound d did not described correctly. Please provide yield and spectral data for compound d.

Author Response

Dear reviewer, thank you for your good suggestions.

Point 1: However, these new compounds were not well characterized. The authors did not provide 13C NMR spectra for new compounds. I recommend preparation of supporting materials with copies of 1H and 13C NMR spectra of synthesized compounds.

Response 1: Supporting materials with copies of 1H NMR, 13C NMR and MS spectra of synthesized compounds have been provided in a PDF file (please see the attachment) and the date has also been added into the revised manuscript (please see the revised manuscript).

Point 2: Line 63-67: Description on figure 2. is not correct. During this study the authors did not performed the ring synthesis. Design strategies could be described as introducing of rigid substituent.

Response 2: ‘ring formation’ was changed to ‘rigidization‘ in figure 2 and the design strategies were described as the following: The flexible diisobutylamino group was replaced with the rigid 3, 5-dimethylpiperidinyl group to optimize the space steric effect of substituent and NH of the phenyl urea group was further replaced with CH to test whether the phenyl urea group is essential to IDO1 potency.

Point 3: Line 71: The synthesis of compound b is nucleophilic aromatic substitution!

Response 3: The synthesis of compound b was described as following: Under basic condition, compound b was obtained through a nucleophilic aromatic substitution reaction with 3,5-dimethylpiperidine.

Point 4: Line 71: The synthesis of compound c is not well known procedure. Please provide reference for this transformation.

Response 4: Reference 14 was provided in the revised manuscript for the transformation.

Point 5: Line 84 and 85: Please provide yields for the step iv and v.

Response 5: The crude product d we got in step iv by preliminary purification of the residue was used directly in step v without further purification. Therefore, the yields for the step iv and v cannot be calculated, respectively. In the revised manuscript, we have calculated the total yield of the two steps (iv and v) and the yield is 32.5%.

Point 6:  Line 241-249: The preparation and characterization of compound d did not described correctly. Please provide yield and spectral data for compound d.

Response 6: The description of the preparation of compound d was revised in procedure 3.1.5 of the manuscript. The crude product d was used directly in step v without further purification. Therefore, the spectral data for compound d was not provided in the manuscript.

Reviewer 2 Report

The article addresses a very relevant issue in the field of design and synthesis of compounds with anti-tumor potential. The chemical synthesis part of this article is interesting; the number of compounds makes it possible to evaluate some structure-activity relationships. However, the article presents some ambiguities that require some clarification.

  • In the abstract part is presented the inhibitory activity of compounds against IDO1 but in article is evaluate also the activity against TDO. The role of this enzyme and the relevance of the evaluation are not described. Please do this and comment the results.
  • The design strategy for the synthesis is based on the BMS-E30 structure. Data regarding the structure and IDO1 inhibitory activity for this compound are supposed to be described in reference 1. Reference 1 contains data for compound BMS-986205. I did not find data for BMS-E30. Please provide the correct reference.
  • The binding mode of BMS-E30 to the IDO1 enzyme is supposed to be similar compared to compound BMS -978587. Please provide than the structure of the BMS -978587 . It can be useful maybe to compare the binding mode of the selected lead compound with this one
  • if is possible, please provide some HNMR and MS spectra

Author Response

Dear reviewer, thank you for your good suggestions.

Point 1: In the abstract part is presented the inhibitory activity of compounds against IDO1 but in article is evaluate also the activity against TDO. The role of this enzyme and the relevance of the evaluation are not described. Please do this and comment the results.

Response 1: The description of the evaluation of TDO inhibitory activity of the synthesized compounds was added into the abstract part and the result that none of them showed TDO inhibitory was also presented in the abstract part. In this article, we tried to design selective IDO1 inhibitors and the evaluation of TDO inhibitory activity was performed to test whether the phenyl urea derivatives were selective IDO1 inhibitors. Therefore, the role of TDO was not described in the article. The comment on the results was also added into the part 2.3 of the revised manuscript.

Point 2: The design strategy for the synthesis is based on the BMS-E30 structure. Data regarding the structure and IDO1 inhibitory activity for this compound are supposed to be described in reference 1. Reference 1 contains data for compound BMS-986205. I did not find data for BMS-E30. Please provide the correct reference.

Response 2: Data regarding the structure and IDO1 inhibitory activity for compound BMS-E30 was described in the reference 11: Balog, J. A.; Huang, A.; Chen, B.; Chen, L.; Shan, W. IDO inhibitors, WO 2014150646 (A1), 25 Sept, 2014. The structure information was described on page 75 and the IDO1 kynurenine assay with human IDO1/HEK 293 cells was described on page 92.

Point 3: The binding mode of BMS-E30 to the IDO1 enzyme is supposed to be similar compared to compound BMS-978587. Please provide than the structure of the BMS -978587. It can be useful maybe to compare the binding mode of the selected lead compound with this one.

Response 3: The structure of the BMS-978587 was added into Figure 2.

Point 4: If is possible, please provide some HNMR and MS spectra.

Response 4: Supporting materials with copies of 1H, 13C and MS spectra of synthesized compounds have been provided in a PDF file (please see the attachment).

Round 2

Reviewer 1 Report

I read revised manuscript under title:

 “ Design, Synthesis and Biological Evaluation of 2 Phenyl Urea Derivatives as IDO1 Inhibitors“

and I can say that the authors have provided an adequate response on my comments and suggestions. Overall, I propose acceptation of revised manuscript in present form.